# NEAR-BLACK-BOX ADVERSARIAL ATTACKS ON GRAPH NEURAL NETWORKS AS AN INFLUENCE MAXIMIZATION PROBLEM

## ABSTRACT

Graph neural networks (GNNs) have attracted increasing interests. With broad deployments of GNNs in real-world applications, there is an urgent need for understanding the robustness of GNNs under adversarial attacks, especially in realistic setups. In this work, we study the problem of attacking GNNs in a restricted near-black-box setup, by perturbing the features of a small set of nodes, with no access to model parameters and model predictions. Our formal analysis draws a connection between this type of attacks and an influence maximization problem on the graph. This connection not only enhances our understanding on the problem of adversarial attack on GNNs, but also allows us to propose a group of effective near-black-box attack strategies. Our experiments verify that the proposed strategies significantly degrade the performance of three popular GNN models and outperform baseline adversarial attack strategies.

## 1 INTRODUCTION

There has been a surge of research interest recently in graph neural networks (GNNs) (Wu et al., 2020), a family of deep learning models on graphs, as they have achieved superior performance on various tasks such as traffic forecasting (Yu et al., 2017), social network analysis (Li et al., 2017), and recommender systems (Ying et al., 2018; Fan et al., 2019). Given the successful applications of GNNs in online Web services, there are increasing concerns regarding the robustness of GNNs under adversarial attacks, especially in realistic scenarios. In addition, the research about adversarial attacks on GNNs in turns helps us better understand the intrinsic properties of existing GNN models. Indeed, there have been a line of research investigating various adversarial attack scenarios for GNNs (Zügner et al., 2018; Zügner & Günnemann, 2019; Dai et al., 2018; Bojchevski & Günnemann, 2018; Ma et al., 2020), and many of them have been shown to be, unfortunately, vulnerable in these scenarios. In particular, Ma et al. (2020) examine an extremely restricted near-black-box attack scenario where the attacker has access to neither model parameters nor model predictions, yet they demonstrate that a greedy adversarial attack strategy can significantly degrade GNN performance due to the natural inductive biases of GNN binding to the graph structure. This scenario is motivated by real-world GNN applications on social networks, where attackers are only able to manipulate a limited number of user accounts, and they have no access to the GNN model parameters or predictions for the majority of users.

In this work, we study adversarial attacks on GNNs under the aforementioned near-black-box scenario. Specifically, an attack in this scenario is decomposed into two steps: 1) select a small set of nodes to be perturbed; 2) alter the node features according to domain knowledge up to a per-node budget. The focus of the study lies on the node selection step, so as in Ma et al. (2020). The existing attack strategies, although empirically effective, are largely based on heuristics (Ma et al., 2020). We instead formulate the adversarial attack as an optimization problem to maximize the mis-classification rate over the selected set of nodes, and we carry out formal analysis regarding this optimization problem. The proposed optimization problem is combinatorial and seems hard to solve in its original form. In addition, the mis-classification rate objective involves model parameters which are unknown in the near-black-box setup. We mitigate these difficulties by rewriting the problem and connecting it with influence maximization on a special linear threshold model related to the original graph structure. Inspired by this connection, we show that, under certain distribu-

tional assumptions about the GNN, the expected mis-classification rate is submodular with respect to the selected set of nodes to perturb. The expected mis-classification rate is independent of the model parameters and can be efficiently optimized by a greedy algorithm thanks to its submodularity. Therefore, by specifying concrete distributions, we are able to derive a group of near-black-box attack strategies maximizing the expected mis-classification rate. The connection with influence maximization also provides us nice interpretations regarding the problem of adversarial attack on GNNs.

To empirically verify the effectiveness of the theory, we implement two near-black-box adversarial attack strategies and test them on three popular GNN models, Graph Convoluntioal Network (GCN) (Kipf & Welling, 2016), Graph Attention Network (GAT) (Veličković et al., 2018), and Jumping Knowledge Network (JKNet) (Xu et al., 2018) with common benchmark datasets. Both attack strategies significantly outperform baseline attack strategies in terms of decreasing model accuracy. Finally, we summarize the contributions of our study as follows.

1. We formulate the problem of adversarial attack on GNNs as an optimization problem to maximize the mis-classification rate.
2. We draw a novel connection between the problem of adversarial attacks on GNNs and influence maximization based on a linear threshold model. This connection helps us develop effective and efficient near-black-box adversarial attack strategies and provides interpretations regarding the adversarial attack problem.
3. We implement two variants of the proposed near-black-box attack strategies and empirically demonstrate their effectiveness.

## 2 RELATED WORK

There has been increasing research interest in adversarial attacks on GNNs recently. Detailed expositions of existing literature are made available in a couple of survey papers (Jin et al., 2020; Sun et al., 2018). Given the heterogeneous nature of diverse graph structured data, there are numerous adversarial attack setups for GNN models. Following the taxonomy provided by Jin et al. (2020), the adversarial attack setup can be categorized based on (but not limited to) the machine learning task, the goal of the attack, the phase of the attack, the form of the attack, and the model knowledge that attacker has access to. First, there are two common types of tasks, node-level classification (Zügner et al., 2018; Dai et al., 2018; Wu et al., 2019; Entezari et al., 2020) and graph-level classification (Tang et al., 2020; Dai et al., 2018). The goal of the attack can be changing the predictions of a small and specific set of nodes (targeted attack) (Zügner et al., 2018; Dai et al., 2018) or degrading the overall GNN performance (untargeted attack) (Zügner & Günnemann, 2019; Sun et al., 2019). The attack can happen at the model training phases (poisoning attack) (Zügner & Günnemann, 2019; Sun et al., 2019) or after training completes (evasion attack) (Dai et al., 2018; Chang et al., 2020). The form of the attack could be perturbing the node features (Zügner et al., 2018; Ma et al., 2020) or altering the graph topology (Dai et al., 2018; Sun et al., 2019). Finally, depending on the knowledge (e.g. model parameters, model predictions, features, and labels, etc.) the attacker has access to, the attacks can be roughly categorized into white-box attacks (Xu et al., 2019), grey-box attacks (Zügner et al., 2018; Sun et al., 2019), black-box attacks (Dai et al., 2018; Chang et al., 2020) or near-black-box (Ma et al., 2020). However, it is worth noting that the borders of these three categories are blurry in literature.

The setup of interest in this paper can be categorized as *node-level, untargeted, evasional, and near-black-box* attacks by perturbing the node features. While each setup configuration might find its suitable application scenarios, we believe that near-black-box setups are particularly important as they are associated with many realistic scenarios. Among the existing studies on node-level black-box attacks, most of them (Bojchevski & Günnemann, 2018; Chang et al., 2020; Dai et al., 2018) still allow access to model predictions or some internal representations such as node embeddings. In this paper, we follow the most strict near-black-box setup (Ma et al., 2020) to our knowledge, which prohibits any probing of the model. Compared to Ma et al. (2020), we develop attack strategies by directly analyzing the problem of maximizing mis-classification rate, rather than relying on heuristics.

We remark that there are also plenty of existing works investigating adversarial attacks on non-GNN models (Wang & Gong, 2019; Zhang et al., 2019), which we consider less relevant to this work, and refer the readers to the survey papers (Jin et al., 2020; Sun et al., 2018) for more details.

## 3 PRELIMINARIES

### 3.1 NOTATIONS

We start by introducing notations that will be used across this paper. Suppose we have an attributed graph $G = (V, E, X, y)$, where $V = \{1, 2, \cdots, N\}$ is the set of $N$ nodes, $E \subseteq V \times V$ is the set of edges, $X \in \mathbb{R}^{N \times D}$ is the node feature matrix with $D$-dimensional features, and $y \in \{1, 2, \cdots, K\}^N$ is the node label vector with $K$ classes. We also denote a random walk transition matrix on the graph as $M \in \mathbb{R}^{N \times N}$. For any $1 \leq i, j \leq N$, $M_{ij} = 1/|\mathcal{N}_i|$ if $(i, j) \in E$ or $i = j$, and $M_{ij} = 0$ otherwise. To ease the notation, for any matrix $A \in \mathbb{R}^{D_1 \times D_2}$ in this paper, we refer $A_j$ to the transpose of the $j$-th row of the matrix, i.e., $A_j \in \mathbb{R}^{D_2}$.

We consider a GNN model $f : \mathbb{R}^{N \times D} \rightarrow \mathbb{R}^{N \times K}$ that maps from the node feature matrix $X$ to the output logits of all nodes (denoted as $H \triangleq f(X) \in \mathbb{R}^{N \times K}$). Let $\mathcal{N}_i = \{j \in V \mid (i, j) \in E\} \cup \{i\}$ be the set of neighbors of node $i$, including itself. We assume the GNN $f$ has $L$ layers, with the $l$-th layer ($0 < l < L$) at node $i$ taking the form $H_i^{(l)} = \text{ReLU} \left( \sum_{j \in \mathcal{N}_i} \alpha_{ij} W^{(l)} H_j^{(l-1)} \right)$. $W^{(l)}$ is the learnable weight matrix, $\text{ReLU}(\cdot)$ is an element-wise ReLU activation function, and different GNNs have different normalization terms $\alpha_{ij}$. We also define $H^{(0)} = X$ and $H = H^{(L)} = \sum_{j \in \mathcal{N}_i} \alpha_{ij} W^{(L)} H_j^{(L-1)}$. Later in Section 4, we carry out our analysis on a GCN model with $\alpha_{ij} = 1/|\mathcal{N}_i|$ (Hamilton et al., 2017).

### 3.2 THE NEAR-BLACK-BOX ADVERSARIAL ATTACK SETUP

Next we briefly introduce the near-black-box adversarial attack setup proposed by Ma et al. (2020). The goal of the attack is to perturb the node features of a few carefully selected nodes such that the model performance is maximally degraded. The attack is decomposed into two steps. In the first step, the attacker selects a set of nodes $S \subseteq V$ to be perturbed, under two constraints $|S| \leq r$ and $|\mathcal{N}_i| \leq m, \forall i \in S$ for some $0 < r \ll N$ and $0 < m \ll \max_i |\mathcal{N}_i|$. These two constraints prevent the attacker from manipulating a lot of nodes or very important nodes as measured by the node degree, which makes the setup more realistic. In the second step, the attacker is allowed to add a small constant perturbation $\epsilon \in \mathbb{R}^D$ to each node in $S$, i.e., let the perturbed feature be $X_i' \triangleq X_i + \epsilon$ for $i \in S$. The perturbation vector $\epsilon$ is constructed based on the domain knowledge about the task but without access to the GNN model. For example, if the GNN model facilitates a recommender system for social media, an attacker may hack a handful of carefully selected users and manipulate their demographic features, posts, or browsing trajectories to get more users exposed to certain political content the attacker desires. In practice, the perturbation vector $\epsilon$ can be tailored for different nodes given personalized knowledge about each node. But following Ma et al. (2020), we consider the worst case where no personalization is available.

### 3.3 INFLUENCE MAXIMIZATION ON A LINEAR THRESHOLD MODEL

Given an information/influence diffusion model on a social network, influence maximization is the problem of finding a small seed set of users such that they spread the maximum amount of influence over the network. In a linear threshold model (Kempe et al., 2003), the influence among nodes is characterized by a weighted directed adjacency matrix $I \in \mathbb{R}^{N \times N}$ where $I_{ij} \geq 0$ for each $(i, j) \in E$ and $I_{ij} = 0$ for each $(i, j) \notin E$. Given a seed set of nodes being activated at initial state, the influence passes through the graph to activate other nodes. There is a threshold vector $\eta \in \mathbb{R}^N$ associated with the nodes, indicating the threshold of influence each node must have received from its active neighbors before it becomes activated. In particular, when the influence propagation comes to a stationary point, a node $i$ outside the seed set will be activated if and only if

$$\sum_{j \in \mathcal{N}_i, j \text{ is activated}} I_{ij} \geq \eta_i. \tag{1}$$

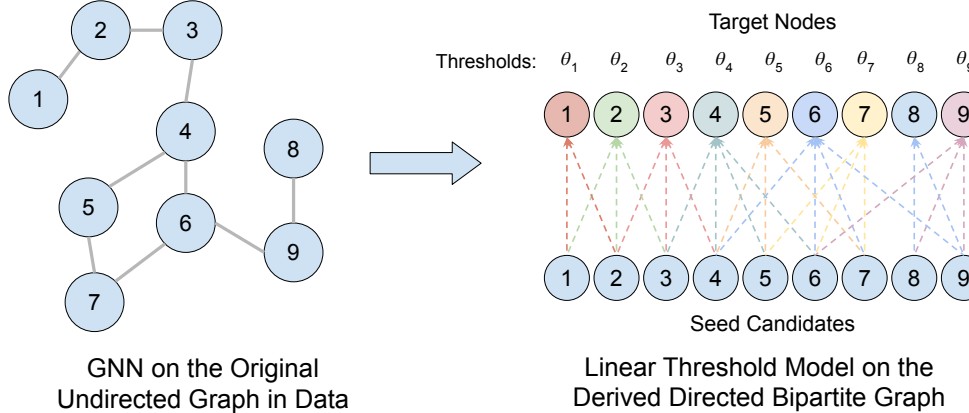

Figure 1: An illustrative example of the linear threshold model on the derived directed bipartite graph. To simplify the visualization, the GNN is assumed to have 1 layer, and therefore the derived directed bipartite graph have links from its zero-th (itself) and first order neighbors in the original graph. For a GNN with $k$ layers, the derived directed bipartite graph will have links from all its $l$-th order neighbors in the original graph, for any $0 \leq l \leq k$. Each target node $i$ has its own threshold $\theta_i$ to be influenced (mis-classified). The edge weight depends on the random walk transition from the seed node to the target node.

## 4    ANALYSIS OF THE ADVERSARIAL ATTACK PROBLEM

In this section, we investigate how to develop adversarial attack strategies under the near-black-box setup stated in Section 3.2 in a principled way.

### 4.1    NODE SELECTION FOR MIS-CLASSIFICATION RATE MAXIMIZATION

Suppose an attacker wants to attack a well-trained $L$-layer GCN model $f$. Following the two-step attack procedure, the attacker first selects a valid node set $S \in C_{r,m} \triangleq \{T \subseteq V \mid |T| \leq r, |\mathcal{N}_i| \leq m, \forall i \in T\}$ for some given constraints $r$ and $m$. Then the constant perturbation $\epsilon$ is added to the feature of each node in $S$, which leads to a perturbed feature matrix $X(S, \epsilon)$. Since our primary interest is the design of the node selection step, we shall omit $\epsilon$ and just write the perturbed feature as $X(S)$ for simplicity. We denote the output logits of the model after perturbation as $H(S) = f(X(S))$. Clearly, $H(\emptyset)$ equals to the matrix of output logits without attack.

In an untargeted attack, the attacker wants the model to make as many mistakes as possible, which is best measured by the mis-classification rate. Therefore we formulate the problem of selecting the node set as as an optimization problem maximizing the mis-classification rate over $S$, with the two constraints quantified by $r, m$:

$$\max_{S \in C_{r,m}} \quad \sum_{j=1}^{N} \mathbb{1} \left[ \max_{k=1,\cdots,K} H_{jk}(S) \neq H_{jy_j}(S) \right], \tag{2}$$

where $\mathbb{1}[\cdot]$ is the indicator function. We drop normalizing constant $1/N$ in mis-classification rate.

At the first glance, the optimization problem (2) is a combinatorial optimization problem with a complicated objective function involving neural networks. In the following section, we demonstrate that, under a simplifying assumption, it can be connected to the influence maximization problem.

### 4.2    CONNECTION TO THE INFLUENCE MAXIMIZATION ON LINEAR THRESHOLD MODEL

We first introduce a simplifying assumption of ReLU that has been widely used to ease the analysis of neural networks (Choromanska et al., 2015; Kawaguchi, 2016), including GCN (Xu et al., 2018).

**Assumption 1** (Xu et al. (2018))**.** *All the ReLU activations activate independently with the same probability, which implies that all paths in the computation graph of the GCN model are independently activated with the same probability of success $\rho$.*

Under Assumption 1, we are able to define $\bar{H}(S) \triangleq \mathbb{E}_{\text{path}}[H(S)]$ for any $S \subseteq V$, where $\mathbb{E}_{\text{path}}[H(S)]$ indicates the expectation of $H(S)$ over the random activations of ReLU functions in the model. Then we can rewrite problem (2) in a form that is similar to the influence maximization objective on a linear threshold model. The influence weight matrix is defined by the $L$-step random walk transition matrix $B \triangleq M^L$. And the threshold for each node is related to the original output logits $\bar{H}(\emptyset)$, the perturbation vector $\epsilon$, and the product of the GCN weights $W \triangleq \rho \cdot \prod_{l=L}^{1} W^{(l)} \in \mathbb{R}^{K \times D}$. Formally, we have the following Proposition 1.

**Proposition 1.** *If we replace $H(\cdot)$ by $\bar{H}(\cdot)$ in problem (2), then we can rewrite the optimization problem as follows,*

$$\max_{S \in C_{r,m}} \quad \sum_{j=1}^{N} \mathbb{1}\left[\sum_{i \in S} B_{ji} > \theta_j\right], \tag{3}$$

*where, for $\hat{k}_j = \operatorname{argmax}_{k=1,\cdots,K} \bar{H}_{jk}(S)$,*

$$\theta_j \triangleq \frac{\bar{H}_{jy_j}(\emptyset) - \bar{H}_{j\hat{k}_j}(\emptyset)}{(W_{\hat{k}_j} - W_{y_j})^T \epsilon}. \tag{4}$$

*In particular, if $\hat{k}_j = y_j$, we define $\theta_j = \infty$.*

**Interpretations of the new objective (3).** The new optimization objective (3) has nice interpretations. The $L$-step random walk transition matrix measures the pairwise influence from input nodes to target nodes in the GCN model and $\sum_{i \in S} B_{ji}$ can be viewed as measuring the influence of nodes in $S$ on a target nodes $j$. In each $\theta_j$, the numerator $\bar{H}_{jy_j}(\emptyset) - \bar{H}_{j\hat{k}_j}(\emptyset)$ can be viewed as the logit margin between the correct class and those wrong classes, which measures the robustness of the prediction on node $j$. The denominator $(W_{\hat{k}_j} - W_{y_j})^T \epsilon$ measures how effective the perturbation is. In combination, $\theta_j$ measures how difficult it is to mis-classify the node $j$ with perturbation $\epsilon$. This new objective nicely separates the influence between nodes and the node-specific robustness.

Note the form of each term inside the summation over $N$ in Eq. (3), $\mathbb{1}\left[\sum_{i \in S} B_{ji} > \theta_j\right]$, is very similar to that of Eq. (1). In fact, the objective (3) can be viewed as the influence maximization objective on a directed bipartite graph derived from the original graph, as shown in Figure 1. The derived bipartite graph has $N$ nodes on both sides (assuming we call them the seed candidate side $\mathcal{S}$ and target node side $\mathcal{T}$), and there are edges pointing from side $\mathcal{S}$ to side $\mathcal{T}$ but not the converse way. The edge weight from the node $i$ on the side $\mathcal{S}$ to the node $j$ on the side $\mathcal{T}$ ($1 \leq i, j \leq N$) is defined as $B_{ji}$. Then it is easy to see that the problem (3) is equivalent to the influence maximization problem on the bipartite graph with the node-specific thresholds being $\theta_j, j = 1, \cdots, N$.

**Two difficulties for solving the problem (3).** While we now have got better interpretations of the original mis-classification rate maximization problem in terms of influence maximization, we still face two major difficulties before we can develop an algorithm to solve the problem. The first difficulty is that we do not known the value of $\theta$ in a near-black-box attack setup as it involves the model parameters. The second difficulty is that, even if $\theta$ is given, influence maximization on the seemingly simple bipartite graph is still NP-hard, as we show in Lemma 1.

**Lemma 1.** *The influence maximization problem on a directed bipartite graph with linear threshold model is NP-hard.*

### 4.3 ASSUMPTIONS ON THE THRESHOLDS

In this section, we mitigate the aforementioned two difficulties by making distribution assumptions on the thresholds $\theta$.

It is well-known that if the threshold $\theta_j$ of each node $j$ is drawn uniformly at random from the interval $[0, 1]$, the expected objective of a general linear threshold model is submodular, which leads to an efficient greedy algorithm that solves the expected influence maximization problem with a performance guarantee (Kempe et al., 2003). In light of this fact regarding the general linear threshold model, we show (in Proposition 2) that a mild assumption on the distribution of $\theta$ will guarantee the expectation of the objective (3) to be submodular, thanks to the simple bipartite structure.

**Proposition 2.** *Suppose the individual thresholds are random variables drawn from some distributions, and the marginal cumulative distribution function of the threshold $\theta_j$ for node $j$ is $F_j$, $j = 1, \cdots, N$. If $F_1, \cdots, F_N$ are individually concave in the domain $[0, +\infty)$, then the expectation of the objective (3),*

$$h(S) \triangleq \mathbb{E}_{\theta_1, \cdots, \theta_N} \sum_{j=1}^{N} \mathbb{1} \left[ \sum_{i \in S} B_{ji} > \theta_j \right], \tag{5}$$

*is submodular.*

Note that here we do not need the thresholds $\theta$ to be independent from each other, and we only require the marginal probability density function of each $\theta_j$ to be non-increasing on the positive region.

Proposition 2 partially addresses the second difficulty. While we still do not have a solution to the original problem (3), we now know that for a wide range of distributions of $\theta$, the expected mis-classification rate is submodular and can be approximated efficiently through a greedy algorithm.

For the first difficulty, we propose to explicitly specify a simple distribution for $\theta$ and optimize the expected mis-classification rate $h(S)$, which no longer involves any model parameters and gives us a near-black-box attack strategy. While this seems to radically deviate from the original optimization objective (3), in the following Section 5, we empirically show that we only need a crude characterization of the distribution of $\theta$ to obtain effective attack strategies.

**Concrete near-black-box attack strategies.** Below we derive two concrete near-black-box attack strategies by specifying the distribution of $\theta$ to be uniform distributions and normal distributions respectively.

**Corollary 1.** *If $a, b > 0$ and $\theta_j \overset{i.i.d.}{\sim} \text{uniform}\,(-b, a)$, then*

$$h(S) = \frac{1}{a+b} \sum_{j=1}^{N} \left( \min(\sum_{i \in S} B_{ji}, a) + b \right), \tag{6}$$

*and $h(S)$ is submodular.*

**Corollary 2.** *If $\sigma > 0$ and $\theta_j \overset{i.i.d.}{\sim} \mathcal{N}(0, \sigma^2)$, then*

$$h(S) = \frac{1}{2} \sum_{j=1}^{N} \left( 1 + erf\left( \frac{\sum_{i \in S} B_{ji}}{\sigma \sqrt{2}} \right) \right), \tag{7}$$

*where $erf(\cdot)$ is the Gauss error function. And $h(S)$ is submodular.*

Corollary 1 and 2 follow directly from Proposition 2 given the cumulative distribution functions of the uniform distribution and the normal distribution as well as the fact that they are concave at the positive region. In particular, Eq. (6) belongs to a well-known submodular function family named the *saturated coverage function* (Lin & Bilmes, 2011; Iyer & Bilmes, 2015). Under assumptions in Corollary 1, the adversarial attack problem reduces to the classic influence maximization problem under the linear threshold model where the thresholds follow uniform distributions.

We name the attack strategies obtained by greedily maximizing the objectives (6) and (7) as **InfMax-Unif** and **InfMax-Norm** respectively. Specifically, each strategy iteratively selects nodes into the set to be perturbed up to a given size. At each iteration, the node, combining with the existing set, that maximizes Eq. (6) or Eq. (7) will be selected.

### 4.4 DISCUSSIONS ON THE APPROXIMATIONS

From problem (3) to our final attack strategies, we have made two major approximations to address the two difficulties that we raised at the end of Section 4.2.

The first approximation is we go from the original optimization problem to its expected version. Note that $\theta$ depends on both the model parameters and the data, which we do not have full access to.

Table 1: Summary of the attack performance in terms of test accuracy (%), the lower the better attack. **Bold** denotes the best performing strategy in each setup. Underline indicates our strategy outperforms all the baseline strategies. Asterisk (*) means the difference between our strategy and the best baseline strategy is statistically significant by a pairwise t-test at significance level 0.05. The error bar ($\pm$) denotes the standard error of the mean by 40 independent trials. The thresholds correspond to the node degree constraint $m$.

| Method | Cora | | | Citeseer | | | Pubmed | | |
|---|---|---|---|---|---|---|---|---|---|
| | JKNet | GCN | GAT | JKNet | GCN | GAT | JKNet | GCN | GAT |
| None | $85.9 \pm 0.1$ | $85.5 \pm 0.2$ | $87.7 \pm 0.2$ | $73.0 \pm 0.2$ | $75.0 \pm 0.2$ | $74.8 \pm 0.2$ | $85.7 \pm 0.1$ | $85.7 \pm 0.1$ | $85.2 \pm 0.1$ |
| | | | | | Threshold 10% | | | | |
| Random | $69.9 \pm 1.1$ | $81.7 \pm 0.3$ | $72.6 \pm 0.6$ | $61.5 \pm 0.9$ | $71.6 \pm 0.2$ | $70.2 \pm 0.5$ | $76.1 \pm 0.6$ | $82.0 \pm 0.3$ | $73.5 \pm 0.3$ |
| Degree | $63.0 \pm 1.4$ | $78.7 \pm 0.4$ | $66.6 \pm 0.7$ | $53.7 \pm 0.9$ | $68.2 \pm 0.3$ | $63.9 \pm 0.5$ | $63.5 \pm 0.9$ | $78.9 \pm 0.5$ | $65.8 \pm 0.7$ |
| Pagerank | $71.7 \pm 0.9$ | $80.1 \pm 0.3$ | $74.2 \pm 0.5$ | $62.3 \pm 0.6$ | $70.0 \pm 0.3$ | $69.7 \pm 0.3$ | $71.8 \pm 0.8$ | $80.2 \pm 0.3$ | $71.2 \pm 0.3$ |
| Betweenness | $63.6 \pm 1.4$ | $80.2 \pm 0.4$ | $64.9 \pm 0.5$ | $54.9 \pm 1.0$ | $70.0 \pm 0.3$ | $65.5 \pm 0.5$ | $67.0 \pm 1.0$ | $78.4 \pm 0.5$ | $62.6 \pm 0.6$ |
| RWCS | $71.8 \pm 0.8$ | $80.3 \pm 0.4$ | $70.8 \pm 0.5$ | $61.9 \pm 0.6$ | $69.9 \pm 0.3$ | $69.4 \pm 0.3$ | $70.8 \pm 0.8$ | $79.7 \pm 0.3$ | $68.9 \pm 0.4$ |
| GC-RWCS | $55.2 \pm 1.5$ | $78.3 \pm 0.5$ | $57.1 \pm 0.6$ | $47.5 \pm 1.0$ | $66.3 \pm 0.5$ | $58.5 \pm 0.6$ | $61.7 \pm 1.1$ | $77.4 \pm 0.6$ | $57.8 \pm 0.8$ |
| InfMax-Unif | $\mathbf{54.3 \pm 1.5}*$ | $\mathbf{77.9 \pm 0.5}*$ | $\mathbf{55.6 \pm 0.6}*$ | $\underline{47.1 \pm 1.0}*$ | $66.2 \pm 0.5$ | $58.4 \pm 0.6$ | $60.0 \pm 1.2*$ | $77.1 \pm 0.7*$ | $57.0 \pm 0.9*$ |
| InfMax-Norm | $\underline{54.6 \pm 1.5}*$ | $78.1 \pm 0.5$ | $56.9 \pm 0.6$ | $\underline{47.1 \pm 1.0}*$ | $\mathbf{65.6 \pm 0.5}*$ | $\mathbf{58.1 \pm 0.6}*$ | $\mathbf{58.8 \pm 1.1}*$ | $\mathbf{76.2 \pm 0.7}*$ | $\mathbf{55.9 \pm 1.0}*$ |
| | | | | | Threshold 30% | | | | |
| Random | $71.5 \pm 1.1$ | $82.1 \pm 0.3$ | $74.1 \pm 0.6$ | $64.0 \pm 0.8$ | $72.4 \pm 0.2$ | $71.7 \pm 0.3$ | $78.0 \pm 0.4$ | $82.4 \pm 0.3$ | $76.0 \pm 0.3$ |
| Degree | $67.5 \pm 1.2$ | $81.0 \pm 0.4$ | $70.4 \pm 0.6$ | $58.4 \pm 1.0$ | $70.5 \pm 0.3$ | $67.7 \pm 0.4$ | $73.2 \pm 0.8$ | $81.1 \pm 0.4$ | $71.0 \pm 0.4$ |
| Pagerank | $79.4 \pm 0.5$ | $82.5 \pm 0.3$ | $82.3 \pm 0.3$ | $70.2 \pm 0.3$ | $72.7 \pm 0.2$ | $73.8 \pm 0.2$ | $79.9 \pm 0.3$ | $82.6 \pm 0.2$ | $79.0 \pm 0.2$ |
| Betweenness | $66.9 \pm 1.3$ | $81.4 \pm 0.3$ | $67.5 \pm 0.5$ | $57.7 \pm 1.0$ | $70.8 \pm 0.3$ | $67.8 \pm 0.5$ | $75.3 \pm 0.5$ | $80.9 \pm 0.4$ | $71.7 \pm 0.4$ |
| RWCS | $79.2 \pm 0.5$ | $82.5 \pm 0.3$ | $82.3 \pm 0.3$ | $69.9 \pm 0.3$ | $72.7 \pm 0.2$ | $73.7 \pm 0.2$ | $78.2 \pm 0.3$ | $81.7 \pm 0.3$ | $77.8 \pm 0.2$ |
| GC-RWCS | $61.9 \pm 1.5$ | $80.2 \pm 0.4$ | $63.2 \pm 0.5$ | $50.6 \pm 1.1$ | $67.8 \pm 0.4$ | $62.1 \pm 0.6$ | $71.1 \pm 0.8$ | $79.9 \pm 0.5$ | $68.8 \pm 0.4$ |
| InfMax-Unif | $\mathbf{58.2 \pm 1.5}*$ | $\mathbf{79.9 \pm 0.4}$ | $\mathbf{59.6 \pm 0.5}*$ | $\underline{49.6 \pm 1.0}*$ | $\mathbf{67.3 \pm 0.5}*$ | $\mathbf{61.2 \pm 0.6}*$ | $\mathbf{69.4 \pm 1.0}*$ | $80.1 \pm 0.5$ | $65.4 \pm 0.5*$ |
| InfMax-Norm | $\underline{58.0 \pm 1.5}*$ | $79.9 \pm 0.5$ | $60.0 \pm 0.5*$ | $\underline{49.5 \pm 1.0}*$ | $67.6 \pm 0.5$ | $61.6 \pm 0.6*$ | $69.6 \pm 1.0*$ | $\mathbf{79.7 \pm 0.5}$ | $\mathbf{65.2 \pm 0.5}*$ |

The first approximation treats them as random, and takes expectation over $\theta$, which integrates out the randomness in data and the model training process. And the resulted expected objective function $h(S)$ is submodular under the conditions in Proposition 2. A natural question regarding this approximation is how does the mis-classification rate (3) concentrate around its expectation (5)? If $\theta$ are independent, the indicator variables in (5) are also independent, and it is easy to show the mis-classification rate is well-concentrated for a large graph size $N$ through Hoeffding's inequality. However, the independence assumption is unrealistic in the case of GNN as the predictions of adjacent nodes should be correlated. Further note that $\theta$ can be written in terms of linear combinations of node features. With extra assumptions on the node features and the graph structure, one may be able to carry out finer analysis on the covariance of $\theta$, and thus how well the mis-classification rate concentrates. We leave this analysis for future work.

The second approximation is that we further specify simple distributions of $\theta$, which highly likely deviate much from the real distribution. On one hand, our superior empirical results shown in Section 5 suggest that these simple strategies are practical enough for some applications. On the other hand, this leaves room for further improvement in real-world scenarios if we have more knowledge regarding the distribution of $\theta$. For example, if an attacker has a very limited number of API calls to access the model predictions, these calls are probably not enough to train a reinforcement-learning-based attack strategies but they can be effectively used to better estimate the distribution of $\theta$.

## 5 EXPERIMENTS

In this section, we first empirically evaluate the performance of the proposed attack strategies, InfMax-Unif and InfMax-Norm, against several baseline attack strategies. We also visualize the distributions of $\theta$ to gain a better understanding of the approximations we made.

### 5.1 ATTACK STRATEGIES FOR COMPARISON

**Implementation of InfMax-Unif and InfMax-Norm.** For the proposed InfMax-Unif and InfMax-Norm, there are two hyper-parameters respectively to be specified. Recall $B = M^L$, the first hyper-parameter for both method is $L$. We set $L = 4$ following RWCS and GC-RWCS. We note that, for the attack strategies to be effective in practice, the hyper-parameter $L$ does not have to be the same as the number of layers of the GNN being attacked, as we will show in the experiments. For InfMax-Unif, there are two additional distribution hyper-parameters $a, b$. However, $b$ does not influence the selection of nodes so we only need to specify $a$. For InfMax-Norm, we need to specify the distribution parameter $\sigma$. We fix $a = 0.01$ and $\sigma = 0.01$ across all the experiment setups.

Theoretically, the optimal choice of $a$ or $\sigma$ should depend on the perturbation vector $\epsilon$ as well as the dataset. However, we find the proposed InfMax-Unif and InfMax-Norm strategies are fairly robust with respect to the choice of $a$ or $\sigma$ (see the sensitivity analysis in Appendix A.3).

**Baseline strategies.** We compare with five baseline strategies, **Degree**, **Betweenness**, **PageRank**, Random Walk Column Sum (**RWCS**), and Greedily-Corrected RWCS (**GC-RWCS**).

The first three strategies, as suggested by their names, correspond to three node centrality scores. These strategies select nodes with the highest node centrality scores subject to the constraint $C_{r,m}$.

RWCS and GC-RWCS are two near-black-box attack strategies proposed by Ma et al. (2020). RWCS is derived by maximizing the cross-entropy classification loss but with certain approximations. In practice, RWCS has a simple form: selects nodes with highest importance scores defined as $I(i) = \sum_{j=1}^{N} [M^L]_{ji}$ (recall that $M$ is the random walk transition matrix). We set the hyper-parameter $L = 4$ following Ma et al. (2020). GC-RWCS further applies a few heuristics on top of RWCS to achieve better mis-classification rate. Specifically, it dynamically updates the RWCS importantce score based on a heuristic. It also removes a local neighborhood of the selected node after selecting each node. In the experiment, we set the hyper-parameters of GC-RWCS $L = 4$, $l = 30$, and $k = 1$ as suggested in their original paper. Interestingly, RWCS can be viewed as a special case of InfMax-Unif if we set $a = \infty$ (or large enough). And GC-RWCS without removing the local neighborhood step can also be viewed a modified version of InfMax-Unif.

## 5.2 THE ATTACK EXPERIMENT

**Experiment setup.** We follow exactly the same experiment setup in Ma et al. (2020) except for that we further include the GAT model (Veličković et al., 2018). So we only briefly introduce the setup here due to the page limit, and refer to Appendix A.2 and Ma et al. (2020) for more details. We test attack strategies on 3 popular GNN models, (2-layer) GCN (Kipf & Welling, 2016), (2-layer) GAT (Veličković et al., 2018), and (7-layer) JK-Net (Xu et al., 2018), for 3 public benchmark datasets, Cora, Citeseer and Pubmed (Sen et al., 2008). We apply the attack strategies following the two-step procedure stated in Section 3.2. For the node selection step, we limit the number of nodes to be attacked, $r$, as 1% of the graph size for each dataset. We test on two setups of the node degree threshold, $m$, by setting it equal to the lowest degree of the top 10% and 30% nodes respectively. For the feature perturbation step, we follow the same way as in Ma et al. (2020) to construct the constant perturbation vector $\epsilon$.

**Experiment results.** We provide the attack experiment results in Table 1. We show the model accuracy after applying each attack strategy in each dataset and model combination, the lower the better. We also include the model accuracy without attack (**None**) and with an attack under random node selection (**Random**) for reference.

As can be seen in Table 1, both the proposed attack strategies achieve better attack performance than all baselines on all but one setups, out of the 18 setups in total. And most of the differences are statistically significant. We highlight that, compared to the strongest baseline, GC-RWCS, our methods have fewer hyper-parameters and better interpretation. In addition, the neighbor-removal heuristic also contributes to the performance of GC-RWCS method, while our methods outperform GC-RWCS without such additional heuristics.

## 5.3 VISUALIZING THE DISTRIBUTIONS OF $\theta$

We also empirically investigate the distributions of $\theta$ to see how likely their PDFs are non-increasing on the positive domain. In particular, given the parameters of a well-trained GCN, we are able to approximately calculate $\theta$ with Eq. (4)[1]. We train a GCN on Cora and get one set of $\theta$. We repeat this process with 1000 independent model initializations and get 1000 sets of $\theta$. Then we can visualize a histogram over the 1000 values of $\theta_j$ for each node $j$. In Figure 2, we show the histograms of 3 randomly selected nodes. We show the histograms of more randomly selected nodes in Appendix A.4. As can be seen from the histograms, in most cases the empirical probability

---

[1]We can only do it approximately because we do not know $\rho$. For the visualization, we just set $\rho = 1$.

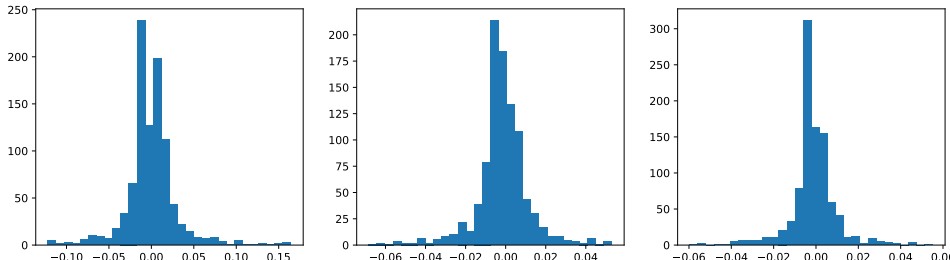

Figure 2: Each figure shows a histogram of $\theta_j$ for a fixed node $j$ over 1000 independent trials of GCN on Cora. The 3 nodes are randomly selected from the union of the validation set and test set.

density decreases when $\theta_j > 0$, which is the assumption required for the expected mis-classification rate to be submodular in Proposition 2.

## 6 CONCLUSION

We present a formal analysis of near-black-box attacks on Graph Neural Networks, formulated as the problem of mis-classification rate maximization. By establishing a novel connection between the original optimization problem to an influence maximization problem upon a linear threshold model, we develop a group of efficient and effective near-black-box attack strategies with nice interpretations. Extensive empirical results demonstrate the effectiveness of the proposed strategies, which outperform state-of-the-art attacking strategies on multiple types of GNNs. In future work, we plan to explore how to perturb the graph structure under this near-black-box setup, as well as how to perturb node features under extra constraints (e.g. binary or nonnegative).

ACKNOWLEDGEMENT

We would like to thank the anonymous reviewers for their detailed comments and suggestions, which help significantly improve this paper.

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

# A  APPENDIX

## A.1  PROOFS

We first give a more precise and restated version (Assumption 2) of Assumption 1, and introduce Lemma 2 about GCN, which is proved by Xu et al. (2018).

**Assumption 2** (Xu et al. (2018) Restated.). *Recall that a ReLU function can be written as*

$$ReLU(x) = x \cdot \mathbb{1}\left[x > 0\right].$$

*Suppose there are $R$ ReLU functions in the GCN model and we index them with $i = 1, 2, \cdots, R$. This assumption assumes that the $i$-th ReLU functions, for $i = 1, 2, \cdots, R$, is replaced by the following function,*

$$ReLU_i(x) = x \cdot z_i,$$

*where $z_1, z_2, \cdots, z_R \overset{i.i.d.}{\sim} Bernoulli(\gamma)$.*

*This assumption implies that all paths in the computation graph of an $L$-layer GCN model are independently activated with the same probability $\rho = \gamma^L$.*

**Lemma 2** (Xu et al. (2018).). *Given an $L$-layer GCN, under Assumption 1, for any node $i, j \in V$,*

$$\mathbb{E}_{path}\left[\frac{\partial H_j}{\partial X_i}\right] = \rho[M^L]_{ji} \cdot \left(\prod_{l=L}^{1} W^{(l)}\right), \tag{8}$$

*where $M \in \mathbb{R}^{N \times N}$ is the random walk transition matrix, i.e., for any $1 \le i, j \le N$, $M_{ij} = 1/|\mathcal{N}_i|$ if $(i, j) \in E$ or $i = j$, and $M_{ij} = 0$ otherwise.*

**Proof for Proposition 1.**

*Proof.* Recall that $\bar{H}(S) = \mathbb{E}_{path}[H(S)] = \mathbb{E}_{path}[f(X(S))]$. We first show $\bar{H}(S)$ is a linear function of $X(S)$, which suffices to show that, for any $i \in V$ and $1 \le l \le L$, $\mathbb{E}_{path}[H_i^{(l)}(S)]$ is a linear function of $\mathbb{E}_{path}[H^{(l-1)}(S)]$. When $l = L$,

$$\mathbb{E}_{path}[H_i^{(L)}(S)] = \sum_{j \in \mathcal{N}_i} \alpha_{ij} W^{(L)} \mathbb{E}_{path}[H_j^{(L-1)}(S)],$$

so the statement holds. When $1 \le l < L$, under Assumption 1, suppose each ReLU activates independently with probability $p$.

$$\mathbb{E}_{path} H_i^{(l)} = \mathbb{E}_{path}\left[\sigma\left(\sum_{j \in \mathcal{N}_i} \alpha_{ij} W^{(l)} H_j^{(l-1)}\right)\right]$$
$$= p \sum_{j \in \mathcal{N}_i} \alpha_{ij} W^{(L)} \mathbb{E}_{path}[H_j^{(l-1)}(S)],$$

so the statement also holds. Therefore $\bar{H}(S)$ is a linear function of $X(S)$. In particular, $\mathbb{E}_{path}[H] = \bar{H}(\emptyset)$ is a linear function of $X$.

We know that $X_i(S) = X_i + \epsilon$ for $i \in S$ and $X_i(S) = X_i$ for $i \notin S$. And by Lemma 2, we can rewrite $\bar{H}(S)$ in terms of $\bar{H}(\emptyset)$ and $\epsilon$. For any $j \in V$,

$$\bar{H}_j(S) = \bar{H}_j(\emptyset) + \sum_{i \in S} \rho[M^L]_{ji} \cdot \left(\prod_{l=L}^{1} W^{(l)}\right)^T \epsilon.$$

In Section 4.2, we have defined $B = M^L$ and $W = \rho \prod_{l=L}^{1} W^{(l)}$, so

$$\bar{H}_j(S) = \bar{H}_j(\emptyset) + W^T \epsilon \sum_{i \in S} B_{ji}. \tag{9}$$

Now we look at the objective (2). If we replace $H(S)$ with $\bar{H}(S)$ in this objective and plug Eq. (9) into it, then for each $j \in V$, we have

$$\mathbb{1}\left[\max_{k \in \{1, \cdots, K\}} \bar{H}_{jk}(S) \neq \bar{H}_{jy_j}(S)\right]$$

$$=\mathbb{1}\left[\bar{H}_{j\hat{k}_j}(S) > \bar{H}_{jy_j}(S)\right]$$

$$=\mathbb{1}\left[\bar{H}_{j\hat{k}_j}(\emptyset) + W_{\hat{k}_j}^T \epsilon \cdot \sum_{i \in S} B_{ji} > \bar{H}_{jy_j}(\emptyset) + W_{y_j}^T \epsilon \cdot \sum_{i \in S} B_{ji}\right]$$

$$=\mathbb{1}\left[\sum_{i \in S} B_{ji} > \frac{\bar{H}_{jy_j}(\emptyset) - \bar{H}_{j\hat{k}_j}(\emptyset)}{(W_{\hat{k}_j} - W_{y_j})^T \epsilon}\right]$$

$$=\mathbb{1}\left[\sum_{i \in S} B_{ji} > \theta_j\right],$$

where we have defined $\hat{k}_j = \mathrm{argmax}_{k=1,\cdots,K} \bar{H}_{jk}(S)$ and recall the definition of $\theta_j$ in Eq. (4).

Therefore we get the optimization problem (3)

$$\max_{S \in C_{r,m}} \sum_{j=1}^{N} \mathbb{1}\left[\sum_{i \in S} B_{ji} > \theta_j\right].$$

$\square$

**Proof for Lemma 1.**    The proof follows similarly as the proof of Theorem 2.4 in Kempe et al. (2003).

*Proof.* We prove by reducing the NP-complete Set Cover problem to the influence maximization problem on directed bipartite graph with a linear threshold model. The Set Cover problem is defined as following. Suppose we have a ground set $U = \{u_1, u_2, \cdots, u_n\}$ and a group of $m$ subsets of $U$, $S_1, S_2, \cdots, S_m$. The goal is to determine whether there exists $r$ ( $r < n$ and $r < m$) of the subsets whose union equals to $U$.

For any instance of the Set Cover problem, we can construct a bipartite graph with the first side having $m$ nodes (each one corresponding to a given subset of $U$), and the second side having $n$ nodes (each one corresponding to an element of $U$). There are only links going from the the first side to the second side. There will be a link with constant influence score $\alpha > 0$ from a node on the first side to the second side if and only if the corresponding subset contains that element in $U$. Finally the node-specific thresholds of each node on the second side is set as $\alpha/2$. And the influence maximization problem asks to select $r$ nodes on the graph to maximize the number of activated nodes. The Set Cover problem is then solved by deciding if the maximized number of activated nodes on the bipartite graph is greater than $n + r$. $\square$

**Proof for Proposition 2.**

*Proof.* We first show that the expected mis-classifcation rate $h(S)$ can be written in terms of the marginal CDFs of $\theta$.

$$h(S) = \mathbb{E}_{\theta_1, \cdots, \theta_N} \sum_{j=1}^{N} \mathbb{1} \left[ \sum_{i \in S} B_{ji} > \theta_j \right]$$

$$= \sum_{j=1}^{N} \mathbb{E}_{\theta_1, \cdots, \theta_N} \mathbb{1} \left[ \sum_{i \in S} B_{ji} > \theta_j \right]$$

$$= \sum_{j=1}^{N} \mathbb{E}_{\theta_j} \mathbb{1} \left[ \sum_{i \in S} B_{ji} > \theta_j \right]$$

$$= \sum_{j=1}^{N} P_j \left( \sum_{i \in S} B_{ji} > \theta_j \right)$$

$$= \sum_{j=1}^{N} F_j \left( \sum_{i \in S} B_{ji} \right),$$

where $P_j$ is the marginal probability of $\theta_j$.

Since $B_{ji} \geq 0$, so $\sum_{i \in S} B_{ji}$ is a non-decreasing submodular function of $S$ with a lower bound 0. Each CDF $F_j$ is non-decreasing by definition, if it is also individually concave at the domain $[0, +\infty)$, we know $F_j \left( \sum_{i \in S} B_{ji} \right)$ is submodular w.r.t. $S$ and hence $h(S)$ is submodular.

$\square$

## A.2 MORE EXPERIMENT DETAILS

**Definitions of the node centralities.** For each node $i$, the Degree centrality score is defined as $C_D(i) \triangleq \frac{|\mathcal{N}_i|}{N}$; the Betweenness centrality score is defined as $C_B(i) \triangleq \sum_{j \neq i, k \neq i, j < k} \frac{g_{jk}(i)}{g_{jk}}$, where $g_{jk}$ is the number of shortest paths connecting node $j$ and $k$ and $g_{jk}(i)$ is the number of shortest paths that node $i$ is on; the PageRank centrality score is defined as the stationary scores achieved by iteratively updating $PR(i) = \frac{1-\alpha}{N} + \alpha \sum_{j \in \mathcal{N}_i} \frac{PR(j)}{|\mathcal{N}_j|}$ and we set the hyper-parameter $\alpha = 0.85$.

**Detailed descriptions of GC-RWCS.** GC-RWCS further applies a few heuristics on top of RWCS to achieve better mis-classification rate. Specifically, it iteratively selects nodes one by one up to $r$ nodes, based on a dynamic importance score, i.e., $I_t(i) = \sum_{j=1}^{N} [Q_t]_{ji}$ for the $t$-th iteration. $Q_t \in \{0, 1\}^{N \times N}$ is a binary matrix that is dynamically updated over $t$. At the initial iteration, $Q_1$ is obtained by binarizing $M^L$, assigning 1 to the top $l$ nonzero entries in each row of $M^L$ and 0 to other entries. For $t > 1$, suppose the node $i$ is selected at the $t - 1$ iteration, then $Q_t$ is obtained from $Q_{t-1}$ by setting to zero for all the rows where the elements of the $i$-th column is 1 in $Q_{t-1}$. GC-RWCS also applies another heuristic that, after each iteration, remove the $k$-hop neighbors of the selected node from the candidate set in the subsequent iterations. In the experiment, we set the hyper-parameters of GC-RWCS $L = 4$, $l = 30$, and $k = 1$ as suggested in their original paper. The iterative-selection process in GC-RWCS (without removing the $k$-hop neighbors) gives equivalent results as InfMax-Unif if we replace the matrix $B$ in InfMax-Unif by $Q_1$ and set $a = 1$.

**More details for the experiment setup.** We randomly split each dataset by 60%, 20% and 20% as the training, validation, and test sets and run 40 independent trials for each model and dataset combination. We apply the attack strategies following the two-step procedure stated in Section 3.2. For the node selection step, we limit the number of nodes to be attacked, $r$, as 1% of the graph size for each dataset. We test on two setups of the node degree threshold, $m$, by setting it equal to the lowest degree of the top 10% and 30% nodes respectively. For the feature perturbation step, we follow the same way as in Ma et al. (2020) to construct the constant perturbation vector $\epsilon$. Ideally, the perturbation vector should be designed according to domain knowledge about the task in real-world scenario. For the experiments on benchmark datasets where we do not know the semantic meaning of the features, we simulate the domain knowledge by extremely limited information of the gradients due to the lack of semantic meaning of each features in benchmark datasets. The gradients

are only used to select important features and the sign of perturbation rather than the magnitude. We construct the $\epsilon_j \in \mathbb{R}^D$

$$\epsilon_j = \begin{cases} \lambda \cdot \text{sign}(\sum_{i=1}^N \frac{\partial L(H,y)}{\partial X_{i,j}}), & \text{if } j \in \arg \text{top}-J([| \sum_{i=1}^N \frac{\partial L(H,y)}{\partial X_{i,j}} |]_{l=1,2,...,D}), \\ 0, & \text{otherwise}, \end{cases} \quad (10)$$

where $\lambda$ is the perturbation strength and is set to 1; and $J$ is set to 2% of the number of features. The same perturbation vector is added to all selected nodes in $S$.

## A.3 ADDITIONAL EXPERIMENTS

**Attack performance with varying perturbation strengths.** In Figure 3, we demonstrate the attack performances of different attack strategies with varying perturbation strengths. We first observe that the proposed attack strategies with the fixed hyper-parameters ($a = 0.01$ for InfMax-Unif and $\sigma = 0.01$ for InfMax-Norm) outperform all baselines in more cases. It is also worth noting that, as suggested by Eq. (4), the distribution of $\theta$ is dependent on the perturbation $\epsilon$ and hence $\lambda$. In the approximated uniform and normal distributions for InfMax-Unif and InfMax-Norm respectively, the optimal choice of $a$ and $\sigma$ should be dependent on $\lambda$. Intuitively, smaller $\lambda$ makes the $\theta$ have larger variance, so the choice of $a$ and $\sigma$ should also be larger. This is indeed suggested by the results in Figure 3. Recall that, in Section 5.1, we discussed that RWCS can be viewed as a special case of InfMax-Unif with $a = \infty$. And in Figure 3, we observe that RWCS (equivalent to InfMax-Unif with $a = \infty$) sometimes (e.g., for GCN) outperforms InfMax-Unif (with $a = 0.01$) when $\lambda$ is very small. However, we leave further optimization of the hyper-parameters of the proposed strategies to future work.

**Sensitivity analysis of $a$ for InfMax-Unif and $\sigma$ for InfMax-Norm.** In Figure 4, we carry out a sensitivity analysis with resepct to $a$ and $\sigma$ for InfMax-Unif and InfMax-Norm respectively. In Section 5.2, we have fixed $a = 0.01$ and $\sigma = 0.01$ for all experiment settings. Here we vary them from 0.005 to 0.02 and show that the results of the proposed strategies, especially those of the InfMax-Norm, stay relatively stable with varying choices of the hyper-parameters.

**Targeting on the test set.** In the experiments in Section 5.2, we use the objectives Eq. (6) and Eq. (7) that sum over the whole graph of $N$ nodes, for an untargeted attack assuming the attacker does not know the test set to be evaluated on. If the targeted test set is known, we can adapt Eq. (6) and Eq. (7) to sum on the test set only. In Table 2, we compare the performance of untargeted attacks and the performance of attacks targeting on the test set. As can be seen, when targeting on the test set, the proposed strategies are further improved compared to their untargeted versions.

**Synthetic data experiments.** We further carry out experiments on synthetic datasets to demonstrate that the proposed attack strategies are effective in a pure black-box setting when sufficient domain knowledge regarding the node features is given. Following Ma et al. (2020), we generate the synthetic datasets as follows. First, we generate a Barabási-Albert random graph (Barabási & Albert, 1999) with $N$ nodes and adjacency matrix $A$. Then we generate node features $X \in \mathbb{R}^{N \times D}$ randomly from a multivariate normal distribution with zero mean and covariance $(L_{\text{sym}} + I)^{-1}$ ($L_{\text{sym}}$ is the symmetric normalized graph Laplacian and $I$ is identity matrix; this covariance introduces smoothness over the graph (Li et al., 2019)), and take the absolute values elementwisely. Finally, node labels are generated by $Y = \mathbb{1}[\text{Sigmoid}((A + I)XW) > 0.5]$, where $W \in \mathbb{R}^D$ is a given weight matrix. During the attack process, we assume that the attacker knows a few ($0.2D$) important features with the largest corresponding weights in $W$ but has no access to the trained model. In Tabel 3, we experiment on 5 synthetic graphs generated by different seeds with $N = 3000$ and $D = 10$, and the proposed InfMax-Unif and InfMax-Norm outperform baseline attack strategies.

**Constructing $\epsilon$ based on the training partition only.** To verify the coarse gradient information we use to construct the perturbation vector $\epsilon$ is not sensitive to the set of nodes, we further repeat the experiments in Table 1 with the only difference that, when constructing $\epsilon$ following Eq. (10), we use the average gradients on the training partition only rather than all nodes. The results are shown in Table 4, which are very similar to those in Table 1. This additional study verifies our belief.

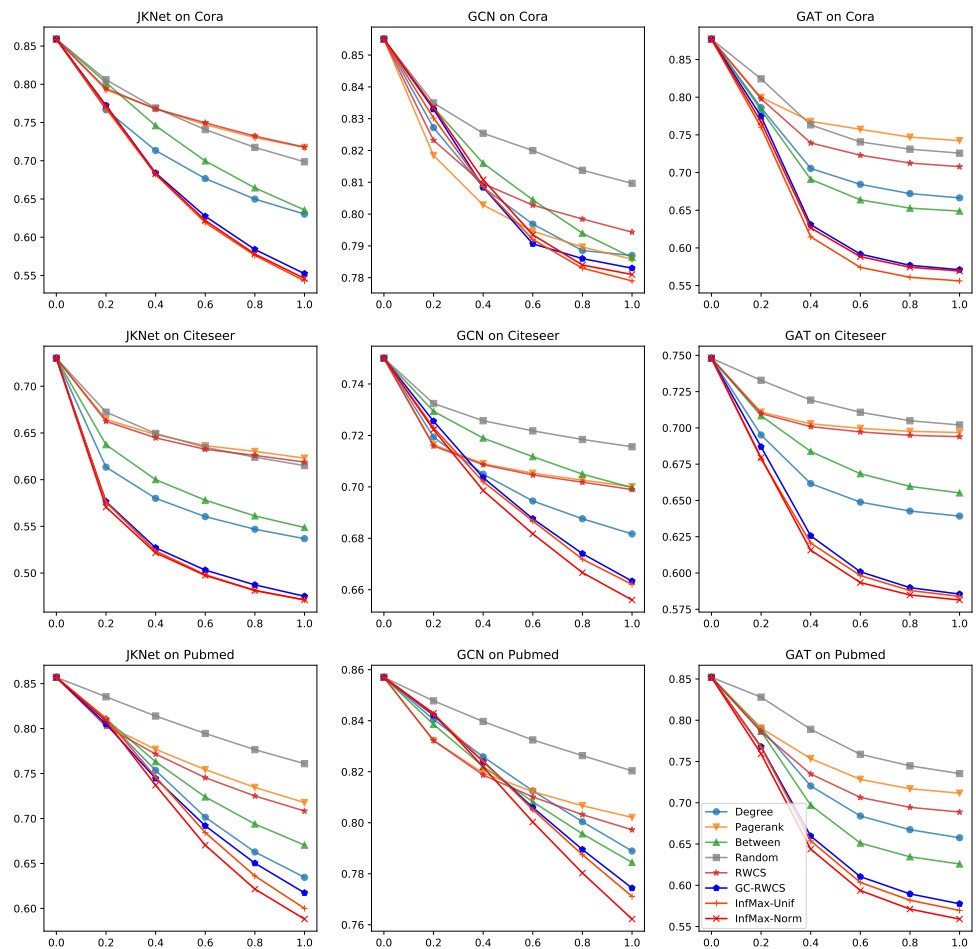

Figure 3: The attack performances with varying perturbation strengths (from 0 to 1). Each figure corresponds to a dataset-model combination. The x-axis indicates the value of $\lambda$ and the y-axis indicates the classification accuracy after attack. The threshold is set as 10% and all other experiment setups are the same as those in Section 5.2.

Table 2: The test accuracy (%) of models after untargeted attacks (U) vs attacks targeting on the test set (T). The experiment setups are the same as those in the Section 5.2.

| Dataset | Cora | | | Citeseer | | | Pubmed | | |
|---|---|---|---|---|---|---|---|---|---|
| Method | JKNet | GCN | GAT | JKNet | GCN | GAT | JKNet | GCN | GAT |
| Threshold 10% | | | | | | | | | |
| InfMax-Unif (U) | 54.3±1.5 | 77.9±0.5 | 55.6±0.6 | 47.1±1.0 | 66.2±0.5 | 58.4±0.6 | 60.0±1.2 | 77.1±0.7 | 57.0±0.9 |
| InfMax-Norm (U) | 54.6±1.5 | 78.1±0.5 | 56.9±0.6 | 47.1±1.0 | 65.6±0.5 | 58.1±0.6 | 58.8±1.1 | 76.2±0.7 | 55.9±1.0 |
| InfMax-Unif (T) | 53.2±1.5 | 77.1±0.7 | 54.2±0.6 | 46.3±1.0 | 65.4±0.5 | 57.7±0.6 | 59.5±1.3 | 77.0±0.7 | 56.0±0.9 |
| InfMax-Norm (T) | 52.4±1.4 | 77.2±0.7 | 53.6±0.6 | 46.1±0.9 | 64.1±0.5 | 56.4±0.7 | 57.9±1.2 | 75.9±0.7 | 54.4±1.0 |
| Threshold 30% | | | | | | | | | |
| InfMax-Unif (U) | 58.2±1.5 | 79.9±0.4 | 59.6±0.5 | 49.6±1.0 | 67.3±0.5 | 61.2±0.6 | 69.4±1.0 | 80.1±0.5 | 65.4±0.5 |
| InfMax-Norm (U) | 58.0±1.5 | 79.9±0.5 | 60.0±0.5 | 49.5±1.0 | 67.6±0.5 | 61.6±0.6 | 69.6±1.0 | 79.7±0.5 | 65.2±0.5 |
| InfMax-Unif (T) | 55.8±1.4 | 78.9±0.5 | 57.5±0.6 | 48.6±1.0 | 66.6±0.4 | 60.2±0.5 | 67.5±1.0 | 78.5±0.6 | 62.9±0.6 |
| InfMax-Norm (T) | 55.3±1.4 | 78.8±0.5 | 57.0±0.6 | 48.6±0.9 | 65.8±0.5 | 59.2±0.6 | 67.4±0.9 | 77.9±0.6 | 62.7±0.6 |

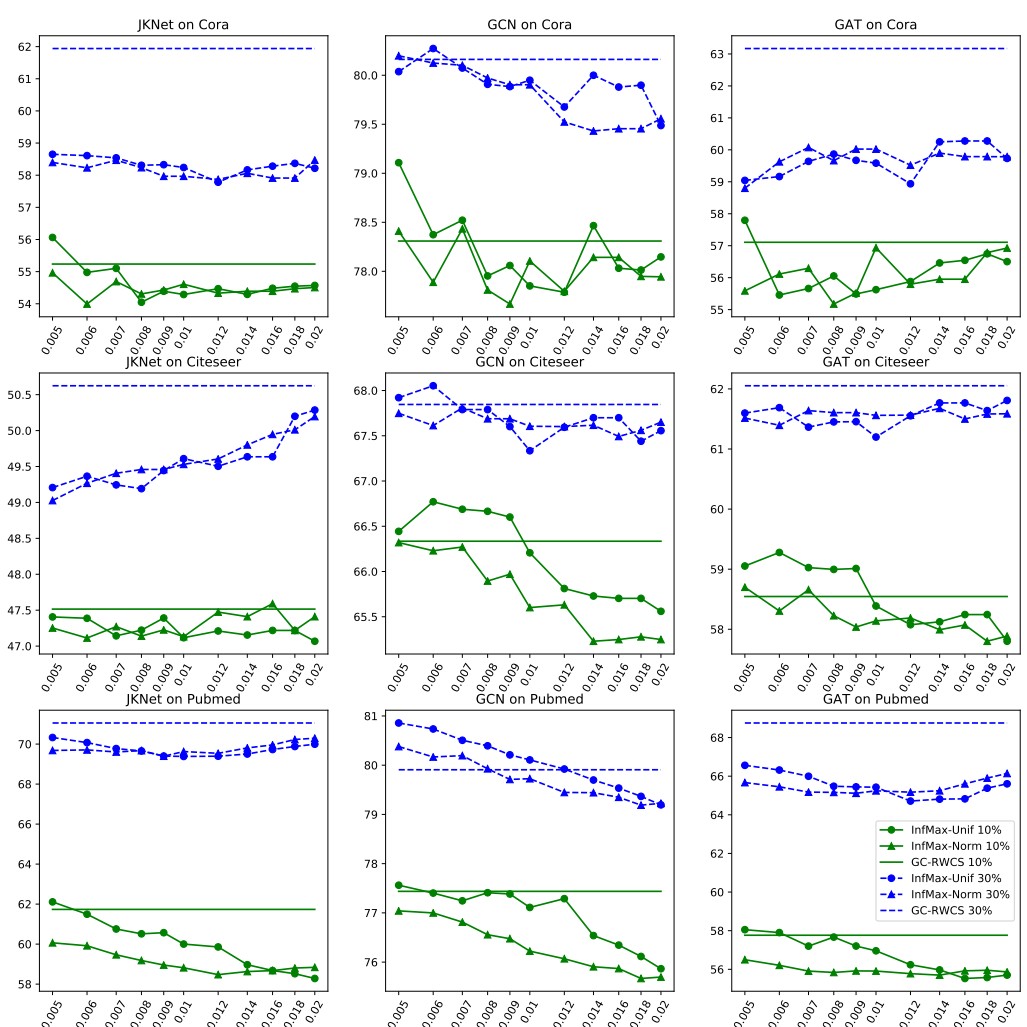

Figure 4: Sensitivity analysis of the hyper-parameters $a$ and $\sigma$. The experiment setups are the same as those in Section 5.2. Each figure corresponds to a dataset-model combination. The x-axis indicates the value of $a$ or $\sigma$ while the y-axis indicates the classification accuracy after attack. The results under the threshold 10% are plotted in green while the results under the threshold 30% are plotted in blue. In addition to the proposed InfMax-Unif and InfMax-Norm, we also plot the results of GC-RWCS as the constant dashed lines for references. The plots are made in log-scale for the x-axis.

Table 3: The test accuracy (%) of GCN model after attacks with 10% threshold on synthetic data. synthetic_0 to synthetic_4 are five synthetic graph generated by different seeds. Other setups are the same as those in the Section 5.2. The notations are the same as those in Table 1.

| Dataset | synthetic_0 | synthetic_1 | synthetic_2 | synthetic_3 | synthetic_4 |
|---|---|---|---|---|---|
| None | 83.9±0.3 | 82.5±0.2 | 82.5±0.2 | 82.0±0.2 | 80.8±0.2 |
| Random | 79.6±0.3 | 78.8±0.2 | 78.3±0.2 | 77.2±0.2 | 77.3±0.2 |
| Degree | 78.2±0.2 | 76.9±0.2 | 75.3±0.3 | 75.5±0.3 | 74.4±0.2 |
| Pagerank | 77.3±0.2 | 77.3±0.3 | 77.2±0.2 | 76.4±0.3 | 75.3±0.2 |
| Betweenness | 78.3±0.2 | 76.6±0.2 | 76.5±0.2 | 76.1±0.2 | 75.3±0.3 |
| RWCS | 78.3±0.2 | 76.6±0.2 | 76.9±0.2 | 76.9±0.2 | 75.1±0.2 |
| GC-RWCS | 77.0±0.2 | 75.6±0.2 | 75.6±0.2 | 75.7±0.2 | 74.6±0.2 |
| InfMax-Unif | 77.4±0.2 | **75.4±0.3** | **74.9±0.3\*** | 74.6±0.2\* | 75.1±0.3 |
| InfMax-Norm | **76.7±0.2\*** | **75.4±0.3** | 75.3±0.3 | **74.5±0.2\*** | **74.3±0.3** |

Table 4: The test accuracy (%) of models when $\epsilon$ is constructed based on the training partition only. The experiment setups are the same as those in the Section 5.2.

| Method | Cora | | | Citeseer | | | Pubmed | | |
|---|---|---|---|---|---|---|---|---|---|
| | JKNetMaxpool | GCN | GAT | JKNetMaxpool | GCN | GAT | JKNetMaxpool | GCN | GAT |
| None | 85.9±0.1 | 85.5±0.2 | 87.7±0.2 | 73.0±0.2 | 75.0±0.2 | 74.8±0.2 | 85.7±0.1 | 85.7±0.1 | 85.2±0.1 |
| | | | | Threshold 10% | | | | | |
| Random | 70.2±1.1 | 81.0±0.3 | 72.4±0.5 | 61.6±0.9 | 71.5±0.3 | 70.6±0.5 | 76.2±0.6 | 82.0±0.3 | 73.8±0.3 |
| Degree | 63.3±1.4 | 78.6±0.4 | 66.7±0.7 | 53.8±1.0 | 68.1±0.3 | 64.7±0.5 | 63.5±1.0 | 78.8±0.5 | 66.2±0.6 |
| Pagerank | 71.9±0.9 | 78.2±0.3 | 74.1±0.5 | 62.4±0.5 | 69.8±0.3 | 70.1±0.3 | 72.0±0.8 | 80.2±0.3 | 71.3±0.3 |
| Betweenness | 63.9±1.4 | 78.6±0.4 | 64.5±0.5 | 55.0±1.0 | 69.7±0.3 | 66.3±0.5 | 67.2±1.0 | 78.4±0.6 | 62.9±0.6 |
| RWCS | 71.9±0.8 | 79.4±0.2 | 70.6±0.5 | 62.0±0.6 | 69.7±0.3 | 69.9±0.3 | 71.0±0.8 | 79.7±0.4 | 69.1±0.4 |
| GC-RWCS | 55.9±1.5 | 77.4±0.6 | 56.9±0.6 | 47.7±1.0 | 66.1±0.5 | 60.1±0.7 | 61.7±1.2 | 77.3±0.7 | 58.3±0.8 |
| InfMax-Unif | **54.8±1.5**\* | **77.3±0.6** | **55.5±0.6**\* | **47.3±1.1**\* | 66.0±0.5 | 59.9±0.8 | 60.0±1.3\* | 77.0±0.7\* | 57.5±0.8\* |
| InfMax-Norm | 55.2±1.6\* | 77.5±0.6 | 56.8±0.6 | 47.3±1.0\* | **65.4±0.6**\* | **59.8±0.7**\* | **58.7±1.2**\* | **76.2±0.8**\* | **56.5±0.9**\* |
| | | | | Threshold 30% | | | | | |
| Random | 71.8±1.1 | 81.5±0.3 | 74.0±0.6 | 64.2±0.8 | 72.4±0.2 | 72.2±0.4 | 78.0±0.4 | 82.4±0.3 | 76.2±0.3 |
| Degree | 67.8±1.2 | 80.2±0.3 | 70.1±0.6 | 58.9±1.1 | 70.5±0.2 | 68.5±0.5 | 73.2±0.8 | 81.0±0.4 | 71.2±0.4 |
| Pagerank | 79.4±0.5 | 82.3±0.1 | 82.4±0.3 | 70.3±0.3 | 72.8±0.2 | 74.0±0.2 | 79.9±0.3 | 82.6±0.2 | 79.0±0.2 |
| Betweenness | 67.2±1.3 | 80.1±0.3 | 67.2±0.5 | 58.1±1.0 | 70.7±0.3 | 68.7±0.4 | 75.4±0.5 | 80.8±0.4 | 71.9±0.4 |
| RWCS | 79.2±0.5 | 82.4±0.2 | 82.4±0.3 | 70.1±0.3 | 72.8±0.2 | 74.0±0.2 | 78.2±0.3 | 81.7±0.3 | 77.9±0.2 |
| GC-RWCS | 62.2±1.5 | 79.8±0.4 | 62.9±0.6 | 51.1±1.1 | 67.9±0.4 | 63.6±0.7 | 71.1±0.8 | 79.8±0.5 | 69.0±0.4 |
| InfMax-Unif | 58.8±1.5\* | **78.9±0.4**\* | **59.4±0.6**\* | 50.2±1.1\* | **67.2±0.5**\* | **62.8±0.7**\* | **69.4±1.0**\* | 80.0±0.5 | 65.9±0.5\* |
| InfMax-Norm | **58.6±1.5**\* | 79.2±0.5\* | 59.8±0.6\* | **50.1±1.1**\* | 67.5±0.5 | 63.1±0.7\* | 69.7±1.0\* | **79.6±0.6** | **65.7±0.5**\* |

## A.4 DISTRIBUTIONS OF $\theta$ ON MORE NODES

Distributions of $\theta$ on more randomly selected nodes are provided in Figure 5. Many examples of the distributions present bell shapes that are close to normal distributions. And it is approximately true that the probability density function is non-increasing at the positive region.

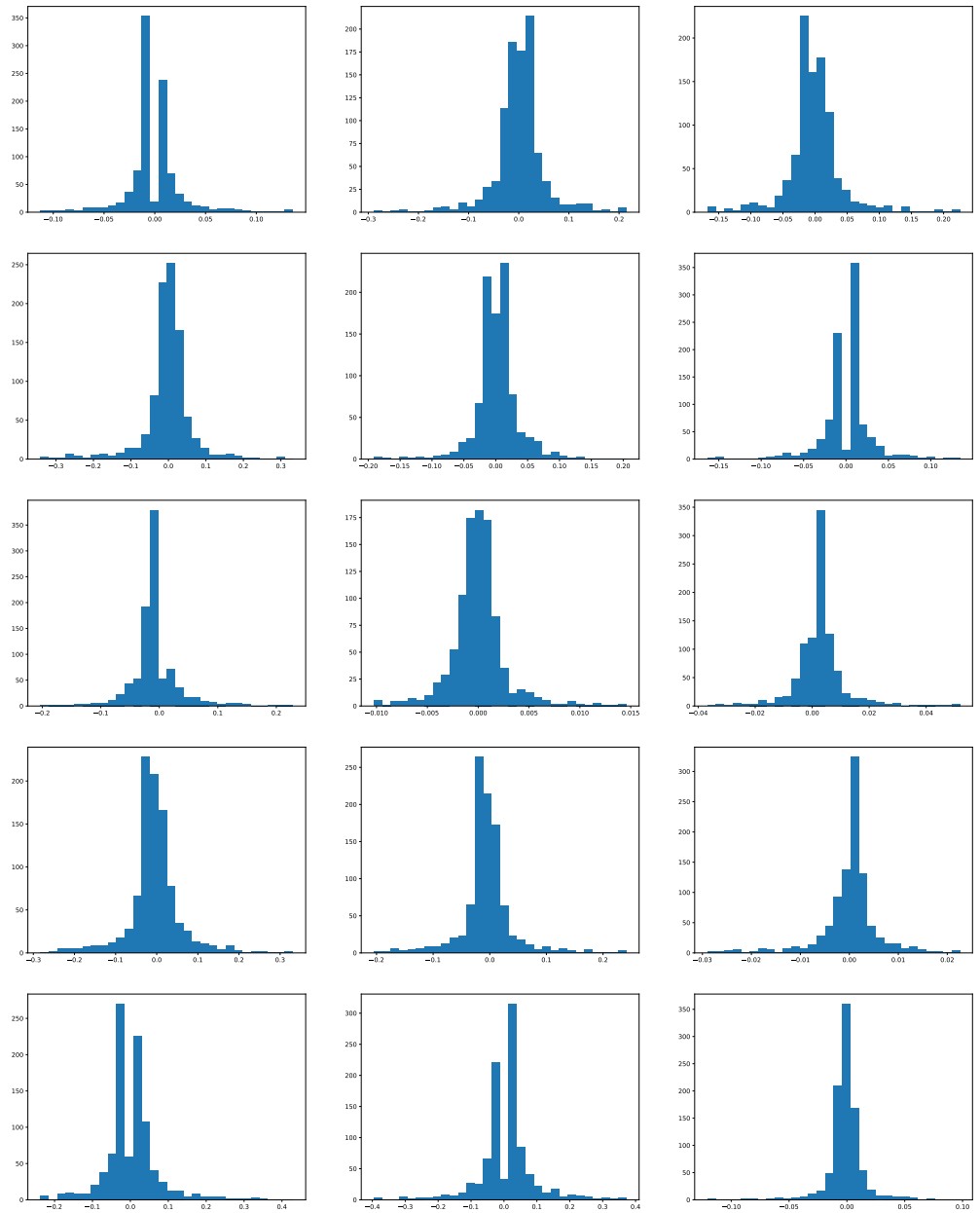

Figure 5: Each figure shows a histogram of $\theta_j$ for a fixed node $j$ over 1000 independent trials of GCN on Cora. The 15 nodes are randomly selected from the union of the validation set and test set.

