# OpenReview forum: "Near-Black-Box Adversarial Attacks on Graph Neural Networks as An Influence Maximization Problem"
_ICLR.cc/2021/Conference — Reject_

### Official Review · AnonReviewer3 · 2020-10-23
**The paper is interesting and the proposed attack method is novel. However, some important details are missing and the threat model is limited.**

**Rating:** 5
**Confidence:** 5

**Review:**

Paper summary: The paper studies the problem of attacking GNNs in a restricted black-box setup, i.e., by perturbing the features of a small set of nodes, with no access to model parameters and model predictions. The authors draw a connection between the restricted attack problem and the influence maximization problem, and then propose several approximation techniques to solve the reformulated attack problem. Experimental results on attacking three GNN models demonstrate the effectiveness of the proposed attack.


Strengths
+The paper is well-written and well organized
+Solving the restricted black-box based on influence maximization is novel and interesting

Weaknesses
-Threat model is limited
-Assumptions are not satisfied
-Some details are unclear
-Missing important references



Detailed comments:

-Threat model is limited.

My first concern is that the proposed attack only focuses on node feature perturbation, while not mentioning structure perturbation. For attacks to graph neural networks, structure perturbation attack is more meaningful and effective. (see Zugner et al., 2018, Dai et al., 2018).

My second concern is that why restricting that the attacker does not know model predictions? What’s the motivation and in what scenarios? From my understanding, even in the black-box attack, an attacker can obtain the model  predictions, e.g., via querying the model using the input nodes.

-Assumptions are not satisfied

The authors assume that theta is from certain simple distributions, in order to make the objective function submodular. However, from the experimental results, such simple distributions are inappropriate.

-Some details are missing

In Equation (4), there is a connection between theta and epsilon.  But in your experimental setup, this connection is lost. Is epsilon necessary in your experiments, as theta is assumed to satisfy some distribution that is irrelevant to epsilon?

How do you obtain the gradient (\partial L / \partial X) in Equation (10) in the black-box setting? What’s the lambda value?

As you use the sign of the gradient and a fixed lambda, all the positive gradients should generate the same value and thus the feature perturbations. In this case, how do you select the feature indexes j as there are many ties in epsilon_j?

In all experimental results, the improvement of the proposed attack over RWCS/GC-RWCS is marginal (less than 3 percent). I think one reason could be that the selected simple distribution for theta largely deviates from the true distribution. Can you plot the true theta distribution, assuming that W is known in advance?

What’s the efficiency of the proposed attack? Is it better than RWCS/GC-RWCS?

-Missing important references
The authors miss the following important works on attacks to graph neural networks:

Wang and Gong, “Attacking Graph-based Classification via Manipulating the Graph Structure”, CCS’19
Wu et al., “Adversarial Examples for Graph Data: Deep Insights into Attack and Defense”, IJCAI’19
Entezari et al., “All You Need Is Low (Rank): Defending Against Adversarial Attacks on Graphs ”, WSDM’20

---

> ### Author Response · Authors · 2020-11-17
> **Response to AnonReviewer3 (Part 1)**
>
> We appreciate the detailed questions and comments by the reviewer and address them below. In particular, we would like to highlight that we provide responses clarifying why the current thread model is meaningful and showing the assumptions on $\theta$ lead to practical attack strategies. We also address the missing details with thorough discussions as well as additional experiments added in the updated draft.
>
> ---
>
> #### Concerns regarding the threat model.
>
> We first explain the motivation of the restricted black-box setup where querying model predictions is limited. A relevant real-world scenario is attacking a GNN-based recommender system on a social network. While the attackers may be able to query the model predictions for a handful of accounts, it is impossible for them to query the model predictions for a massive number of accounts.
>
> [Ma et al. 2020] proposed this setup and had a detailed discussion of the motivations. So in our submission we refer the motivations to that literature as the setup is not our contribution. However, we agree that elaborating the motivations is beneficial for the completeness of this paper and for readers who are not familiar with the literature, and thanks for raising up this question! We added a brief discussion at the end of the first paragraph of Section 1 in our updated draft.
>
> Regarding the perturbation type, we agree that structure perturbation attack is a more interesting problem. However, such an attack is also technically more challenging under this realistic but extremely restricted black-box setup. So we leave this problem as a future work. We also updated our conclusion to include this future direction.
>
> Finally, we would like to highlight that, under the restricted black-box setup, even the problem of node feature perturbation raises difficult technical challenges left open by [Ma et al. 2020]. And this work fills the gap through non-trivial analysis of the problem.
>
> ---
>
> #### Assumptions on $\theta$ distribution
>
> We acknowledge that the uniform and normal assumptions are somewhat oversimplified assumptions. However, we make the following three remarks.
>
> (1) The main goal of the analysis is to develop practically effective adversarial attack strategies. Due to the restricted information available to the attackers, it is inevitable to make certain approximations of the model to derive attack strategies. We highlight that, despite the approximations brought by our assumptions, the proposed strategies are still empirically effective.
>
> (2) For the expected objective to be submodular, we only require the distribution of $\theta$ to be non-increasing on the positive region. The real distribution of $\theta$ seems to align well with this condition according to our visualizations in Figure 2 in Section 5.3 and Figure 5 in Appendix A.4 (Figure 3 in Appendix A.3 in the original draft). Further, many examples of the real distribution of $\theta$ present bell shapes that are close to normal distributions.
>
> (3) While we have made simplifying assumptions, our proposed strategies are still more interpretable and theoretically justified compared to baseline methods. In Section 5.1 and Appendix A.2, we discussed how the baselines RWCS and GC-RWCS can be viewed as special variants of the proposed InfMax-Unif strategy.
>
> ---
>
> (To be continued)

---

> > ### Author Response · Authors · 2020-11-17
> > **Response to AnonReviewer3 (Part 2)**
> >
> > #### Missing details
> >
> > (1) Connection between $\theta$ and $\epsilon$.
> >
> > This is a great question! First we clarify that $\epsilon$ is the perturbation vector to be added to the node features. So it is always needed to complete the attack process in the experiments.
> >
> > Regarding the distribution of $\theta$, it is indeed affected by $\epsilon$. Even with approximations we made in Corollary 1 (uniform) and 2 (normal), they are still relevant to $\epsilon$ through the parameters $a$ or $\sigma$. Ideally, we could optimize the choice of $a$ or $\sigma$ according to $\epsilon$ as the attacker knows the perturbation vector $\epsilon$. However, in this work, we find even simple choices of $a$ or $\sigma$ would already result in effective attack performance, leaving room for further optimizations in future work.
> >
> > An additional evidence that the distribution of $\theta$ is affected by $\epsilon$ lies in the additional experiment (Figure 3 in the Appendix A.3) of different perturbation strength. Recall that the baseline RWCS is a special case of InfMax-Unif with $a=\infty$. Intuitively, when the perturbation strength is small, the parameter $a$ or $\sigma$ should be larger. And, in Figure 3, indeed we find that RWCS outperforms InfMax-Unif (with $a=0.01$) in a few settings and when the perturbation strength is small.
> >
> > (2) The gradient.
> >
> > As explained before Eq. (10), in practice, the construction of the perturbation vectors should be done with the domain knowledge of the task and the semantic meaning of the node features. In our experiments, as there lacks semantic meaning of the features, we are following [Ma et al. 2020] to use very mild information from the average gradient to simulate the domain knowledge of the task to construct the perturbation vector. To verify our methods in a pure black-box setting, we further added a synthetic data experiment in Table 3 in the updated draft, where we are able to know the important features without access to the model.
> >
> > We also added in the updated draft that $\lambda$ is fixed as 1.
> >
> > (3) Selection of the feature indexes j.
> >
> > We clarify that the top feature indexes j are selected in terms of the original average gradients. And the selected few indexes are assigned to a constant $\lambda$ multiplying the sign of the gradient to form a sparse perturbation vector $\epsilon$.
> >
> > (4) Improvement over baselines.
> >
> > We first note that GC-RWCS is a strong baseline. As discussed in the Appendix A.2, it can be viewed as a special variant of the proposed method. We also highlight that the advantages of the proposed strategies over GC-RWCS are three-fold: 1) performance gain, 2) fewer hyper-parameters, and 3) better interpretations.
> >
> > Regarding the distribution of $\theta$, we did plot some real distributions assuming W is known in Figure 2 as well as Figure 5. And many examples of the real distribution of $\theta$ present bell shapes that are close to normal distributions.
> >
> > (5) The efficiency
> > In practice, both the proposed strategies and RWCS/GC-RWCS are very fast. We call them "efficient" in the sense that the solutions are efficient compared to solving the original combinatorial optimization problem.
> >
> > A more detailed analysis is shown here. Both the proposed strategies and RWCS/GC-RWCS need to first compute an L-step random walk probability matrix B through sparse matrix multiplication, which is O(NE), where N is the number of nodes and E is the number of edges. For RWCS, one only needs to have a column sum of B, then take argmax over the sums, which in total is O(N^2). For the proposed strategies and GC-RWCS, we need to do some minor updates on B and do the column sum for each node to be selected. And the operation in each step is still O(N^2). To greedily select r nodes, we would have a time complexity of O(rN^2). Note r << N. Further, in the case when we only target at the test set of, say, S nodes, all the O(N^2) will be reduced to O(NS). We also note that the most of the O(N^2) operations can be made parallel and exploit the sparsity of the matrix B.
> >
> > ---
> >
> > #### Missing references
> >
> > Thanks for the pointers! We have added these references in the updated draft.

---

> > > ### Comment · AnonReviewer3 · 2020-11-24
> > > **Many Details are Still Unclear**
> > >
> > > Thank you for the response.  I am still unclear about several details.
> > >
> > > --Connection between  $\theta$ and $\epsilon$ is still unclear.
> > > The exact connection between $\theta$ and $\epsilon$ is shown in Equation (4). Due to the difficulty to calculate Equation (4),  you require that $\theta$ satisfies some simple distributions associated with $a$ and $\sigma$ (In Corollary 1 and Corollary 2).  You said that $a$ and $\sigma$ depend on $\epsilon$, but in your experiments, the value of $a$, $\sigma$, and $\epsilon$ are independently set.
> > >
> > > -How to obtain the gradient is still unclear.
> > > You claimed that the attacker did not need to query the model. However, when calculating the gradient, you do require the attacker to know the loss (defined in Equation 10) for all nodes, which means the attacker needs to query the model. How did you obtain these values?

---

> > > > ### Author Response · Authors · 2020-11-24
> > > > **Thank you for the follow-up and here are more clarifications.**
> > > >
> > > > Thank you very much for the follow-up and for providing us the opportunity to make further clarifications!
> > > >
> > > > ---
> > > >
> > > > (1) Connection between $\theta$ and $\epsilon$.
> > > >
> > > > We guess the confusion is possibly due to the subtle distinction between the $a$ (or similarly for $\sigma$) as the **distribution parameter** of $\theta$ and the $a$ as the **hyper-parameter** of the derived attack strategy.
> > > >
> > > > This distinction is perhaps easier explained through an analogy to fitting data from Gaussian mixtures. Suppose we are given a dataset generated by a mixture of $k=3$ Gaussian distributions. However, the ground-truth number of mixtures $k$ is not revealed. Therefore we have to treat $k$ as a hyper-parameter and, in practice, we may find fitting a Gaussian mixture model with $k$ set to 5 still achieves good performance.
> > > >
> > > > In our case, the $a$ as the distribution parameter is relevant to $\epsilon$, analogous to the ground-truth $k=3$. However, the exact dependence $a(\epsilon)$ is unknown due to the limited attacker knowledge. In our attack strategy, we treat $a$ as a hyper-parameter (analogous to the hyper-parameter $k=5$) and set it as a fixed value.
> > > >
> > > > We note that **it is possible (though non-trivial) to further optimize the hyper-parameter $a$ according to the choice of $\epsilon$**, which should lead to even better attack performance. However, our experiment results demonstrate that even without careful optimization of the hyper-parameter, the proposed attack strategies already achieve decent attack performance. Moreover, we have conducted sensitivity analysis in Figure 3 and Figure 4 in the Appendix A.3 to show that the proposed strategies are fairly robust to the hyper-parameter choice.
> > > >
> > > > We also updated the "Implementation of InfMax-Unif and InfMax-Norm" part in Sec. 5.1 to reflect the clarifications.
> > > >
> > > > ---
> > > >
> > > > (2) Regarding the gradient.
> > > >
> > > > We clarify that the use of gradients is more about the experiment setup than the attack strategy.
> > > >
> > > > In many real-world scenarios, we often know a couple of critical node attributes based on our knowledge of the prediction task. For example, the attribute of "age" might be a critical feature when predicting "income". In these cases, we can construct a perturbation vector $\epsilon$ based on the semantic meaning of the features, without the need to query the model.
> > > >
> > > > For the three benchmark datasets (Cora, Citeseer, and Pubmed), however, we have lost track of the semantic meaning of the features. To construct a meaningful perturbation vector $\epsilon$ for the purpose of experimental evaluation, we use the gradients to *simulate* the domain knowledge of the task. Nevertheless, we would like to highlight that
> > > >
> > > > - **very mild gradient information** is used to construct $\epsilon$: we only use the coarse population-level (rather than node-level) average gradients to identify which few entries of $\epsilon$ to be non-zero, and the magnitudes of these non-zero entries are irrelevant to the gradients;
> > > > - our paper focuses on the node selection step, and all the attack strategies (including ours and all the baselines) use **exactly the same $\epsilon$** so the comparison is fair;
> > > > - we follow exactly the same setup as [Ma et al. 2020], which is the reason we leave the detailed description of this experiment setup in Appendix A.2.
> > > >
> > > > We believe such mild information, i.e., a few important features for the prediction task, is usually available from domain knowledge and thus our simulation is a reasonable proxy of domain knowledge. To verify this, we also further have **an additional synthetic data experiment**. In particular, we generate synthetic node classification data where a couple of important features for the task are known as (model-agnostic) domain knowledge, so that we can construct a constant perturbation vector in the pure black-box setting **without any query to the model**. And we demonstrate that the proposed strategies still outperform the baselines (see Table 3 in Appendix A.3).

---

> > > > > ### Comment · AnonReviewer3 · 2020-11-24
> > > > > **More Details on the Connection and Gradient**
> > > > >
> > > > > (a) I understand that you said $a$ is an implicit function of $\epsilon$. What I am confused is: in your experiment setup, you individually set $a$ and $\epsilon$, then how can you say they are related to each other?
> > > > >
> > > > > (b) In your experiments, how did you exactly obtain the population-level estimated gradients? You first assume that you know the losses, calculate the population-level average gradient, and then select the top-J indexes (following Ma et al.)?

---

> > > > > > ### Author Response · Authors · 2020-11-24
> > > > > > **More explanations.**
> > > > > >
> > > > > > Thank you again for the follow-up!
> > > > > >
> > > > > > ---
> > > > > >
> > > > > > (a)
> > > > > >
> > > > > > For the distribution parameter $a$, it should be implicitly dependent on $\epsilon$ but unknown.
> > > > > >
> > > > > > For the hyper-parameter $a$, if we were to conduct very careful hyper-parameter search on $a$ given $\epsilon$, then for different $\epsilon$, the *optimal hyper-parameter* $a$ should also differ; and hence in this sense, the *optimal* hyper-parameter $a$ is relevant to $\epsilon$.
> > > > > >
> > > > > > In our experiment, we find that not carefully searching the hyper-parameter $a$ also works fine in practice. So we just set it as a fixed value. In this sense, the *fixed hyper-parameter* we set is indeed irrelevant to $\epsilon$.
> > > > > >
> > > > > > ---
> > > > > >
> > > > > > (b)
> > > > > >
> > > > > > Yes, your description is correct. That is the way we simulate the domain knowledge following Ma et al. (2020).
> > > > > >
> > > > > > To further address the potential concern regarding the use of the term "black-box" in a strict sense, we are changing the term from "black-box" to "near-black-box" in our draft (to be updated soon), per the suggestion by AnonReviewer4.

---

### Official Review · AnonReviewer1 · 2020-10-26
**The paper studies an interesting problem but the settings are not very realistic.**

**Rating:** 5
**Confidence:** 4

**Review:**

##########################################################################
Summary:
This paper studies the problem of designing adversarial attacks (on GNN models) that perturb the feature to maximize the misclassified instances. Assuming that the activations are activated independently at random, the paper shows that the attack design can be reduced to the influence maximization problem under the threshold model. The paper identifies several conditions on the threshold that can make the influence maximization problem submodular, thereby making it easy to optimize.  Experiments have been shown that the proposed attack method has higher performance compared to the existing ones.

##########################################################################
I find the paper interesting but lean towards rejection at this point. The main reason is that the assumptions are not realistic, and the contributions are incremental.
##########################################################################

Strength:

The technical proofs seem to be sound, and the relationship between the attack design and influence maximization is an interesting observation.

Weakness:

The entire analysis is based on Assumption 1, but the paper does not provide a formal description of the data flow in the network under such an assumption, which makes it hard to follow the subsequent analysis. In addition, this assumption is very restrictive in the sense that it makes all the activation functions (\sigma) output random numbers, ignoring what has been received from the last layer. For theoretical analysis on general neural networks, such an assumption is somehow minimal (though not realistic, as pointed out in Kawaguchi 2016) to obtain theoretical results, but it is overly strong for designing a practical attack. In particular, assuming Assumption 1 means that we have ignored the structure of GCN, and therefore, it is not appropriate to target this paper for GCN model. In general, I would not think a meaningful attack could be designed without having certain types of prior knowledge.

Another concern is that the technical analysis largely follows Ma 2020 - the proofs and ideas are very similar. From the introduction, the main difference is to adopt Assumption 1 and get theoretical results, which is somehow incremental.

The amount of perturbation is not extensively discussed in the experiments; to me, the strength of perturbation is critical to the attack performance.

Minor issues:

It would better to introduce the goal of the attack along with the attack setup.
Please define \E_path[H(S)] before using it.
Please introduce RWCS and GC_RWCS before using these terms.

---

> ### Author Response · Authors · 2020-11-17
> **Response to AnonReviewer1**
>
> We appreciate the reviewer for the detailed comments and we address your specific concerns in this response. In particular, we would like to highlight that there seems to be some misalignment between our understandings of Assumption 1 and the work of [Ma et al. 2020], as we clarify below. We also added *an additional sensitivity analysis* for the perturbation strength in our updated draft.
>
> ---
>
> #### Concerns regarding assumption 1
>
> Assumption 1 is indeed an oversimplified assumption for neural networks and we agree that one should be careful about the context to properly use this assumption. However, we think the use of this assumption is proper in our case for the following reasons.
>
> (1) We would like to clarify that assumption 1 does NOT ignore the (graph) structure of GCN. It only makes approximations on the ReLU functions. In fact, it has been reported that assumption 1 aligns particularly well with GCN in practice. For evidence, this assumption is used in quite a few existing GNN literatures [1,2,3]. This is also explicitly stated by Keyulu Xu (the author of [1]) in a public comment in this openreview post: https://openreview.net/forum?id=S1ldO2EFPr&noteId=SklIn_C3uB
>
> (2) Even under the simplified assumption 1, the proposed attack strategies already present practical and superior attack effects in the empirical experiments. Relaxing the assumption might lead to even better attack strategies, which would be an interesting future direction to explore.
>
> (3) An attacker usually has limited information about the model and simplifying assumptions are often involved when designing an adversarial attack strategy. In this sense, the task of designing adversarial attacks is even more tolerant to simplified assumptions compared to that of improving classification models.
>
> ---
>
> #### Comparison with [Ma et al. 2020]
>
> First, we would like to clarify that Assumption 1 is also used in [Ma et al. 2020] so this is not the difference between our work and theirs.
>
> Next, while we follow the black-box attack setup and the experiment setup in [Ma et al. 2020], the technical analyses, including the proofs and the proposed attack strategies, are significantly different. And we highlight the main differences below.
>
> (1) Our analysis targets on directly maximizing the mis-classification rate, which is a discrete function; [Ma et al. 2020] focus on the analysis of the smooth classification loss. Our analysis on the discrete function is more challenging and the proofs are very different.
>
> (2) As a result of (1), the attack strategies derived from our theoretical analysis are directly effective on increasing the mis-classification rate. In [Ma et al. 2020], however, the attack strategy (RWCS) derived from their analysis needs to be combined with heuristic tricks (GC-RWCS) to work well in practice. Our proposed attack strategies have less hyper-parameters and are more interpretable than that of [Ma et al. 2020]. In Section 5.1 and Appendix A.2, we also discussed how RWCS and GC-RWCS can be viewed as special variants of the proposed InfMax-Unif.
>
> (3) Our analysis establishes a novel connection between the adversarial attack on the graph neural networks and the well-studied influence maximization problem, which not only obtains stronger results but also opens up many opportunities for follow-up explorations.
>
> ---
>
> #### The strength of the perturbation
>
> Yes, the strength of perturbation does affect the attack performance but we find the relative rank between different attack strategies is pretty stable. We further added a sensitivity analysis with respect to the strength perturbation (see Figure 3 in the Appendix) in the updated draft.
>
> ---
>
> #### Minor Issues
>
> Thanks for pointing out the unclear parts! We updated the draft according to the suggestions.
>
> ---
>
> #### References
>
> [1] Keyulu Xu, , Chengtao Li, Yonglong Tian, Tomohiro Sonobe, Ken-ichi Kawarabayashi, Stefanie Jegelka. Representation Learning on Graphs with Jumping Knowledge Networks. ICML 2018.
>
> [2] Yuexin Wu, Yichong Xu, Aarti Singh, Yiming Yang, Artur Dubrawski. Active Learning Graph Neural Networks via Node Feature Propagation. ArXiv 2019.
>
> [3] Jiaqi Ma, Shuangrui Ding, Qiaozhu Mei. Black-box adversarial attacks on graph neural networks with limited node access. NeurIPS 2020.

---

> > ### Comment · AnonReviewer1 · 2020-11-24
> > **About assumption 1**
> >
> > Thanks for the response.
> >
> > Assumption 1 states that all the ReLU activations activate independently with the same probability. To me, this means that they simply generate random numbers, ignoring the input. Could you give a formal description of this assumption, preferably, with math?

---

> > > ### Author Response · Authors · 2020-11-24
> > > **Thanks for the follow-up and we updated the draft accordingly!**
> > >
> > > Thank you for the follow-up and for the suggestions to further improve our clarity!
> > >
> > > Now in our updated draft, we have restated Assumption 1 as Assumption 2 in Appendix A.2 and made it precise in math.
> > >
> > > At a high level, this assumption does make the **activation** (but NOT the **output**) of ReLU ignore the input. We would like to highlight that this assumption does NOT ignore the structure of GCN, which is why, under this assumption, the derived attack strategies are still effective.

---

> > > > ### Comment · AnonReviewer1 · 2020-11-24
> > > > **It makes better sense now.**
> > > >
> > > > Thanks for the clarification, and now it makes better sense. I am happy to raise my score to 5 but still have some concerns.
> > > >
> > > > ----------------------------------
> > > > While this assumption is less strong, I do not see the reason for adopting it, except for getting theoretical results. In addition, what does it mean by saying that assumption 1 aligns particularly well with GCN in practice? Can you provide experimental evidence in the existing works?
> > > >
> > > > Minor:
> > > >
> > > > Figure 3 is not black-white-print friendly.

---

> > > > > ### Author Response · Authors · 2020-11-24
> > > > > **More elaborations on Assumption 1**
> > > > >
> > > > > Thanks for updating the evaluation. That is encouraging!
> > > > >
> > > > > ---
> > > > >
> > > > > Beyond the theoretical justifications, this assumption also helps us derive practically effective attack strategies. We note that the derivation of Eq. (3) relies on this assumption, which is a critical foundation for the development of the following concrete and practical attack strategies.
> > > > >
> > > > > There are multiple existing works in the GNN literature where the developments of practical techniques rely on assumption 1. For example, [1] developed a skip-connection structure on top of GCN with theoretical justifications relying on assumption 1; [2] derived a principled active learning algorithm for GNNs with assumption 1 as a critical foundation; [3] derived adversarial attack strategies also under assumption 1.
> > > > >
> > > > > Despite the approximation made by assumption 1, the new GNN architectures [1], active learning algorithms [2], and adversarial attack strategies [3 and ours], all demonstrate **effective empirical performances**, which are strong empirical evidence that assumption 1 aligns well with the **practical behavior** of GCN.
> > > > >
> > > > > ---
> > > > >
> > > > > We will change the plot of Figure 3 in the final version of this draft. Thanks for the thorough consideration!
> > > > >
> > > > > ---
> > > > >
> > > > > References
> > > > >
> > > > > [1] Keyulu Xu, Chengtao Li, Yonglong Tian, Tomohiro Sonobe, Ken-ichi Kawarabayashi, Stefanie Jegelka. Representation Learning on Graphs with Jumping Knowledge Networks. ICML 2018.
> > > > >
> > > > > [2] Yuexin Wu, Yichong Xu, Aarti Singh, Yiming Yang, Artur Dubrawski. Active Learning Graph Neural Networks via Node Feature Propagation. ArXiv 2019.
> > > > >
> > > > > [3] Jiaqi Ma, Shuangrui Ding, Qiaozhu Mei. Black-box adversarial attacks on graph neural networks with limited node access. NeurIPS 2020.

---

### Official Review · AnonReviewer4 · 2020-10-29
**Official Blind Review**

**Rating:** 5
**Confidence:** 5

**Review:**

Summary of the paper:
--------------

The authors propose an adversarial attack strategy for graph neural networks based on influence maximization. The attack is (claimed to be) black-box (does not have direct access to the model), evasion-based, and limited to perturbations of the node attributes (i.e. cannot insert / delete edges). The authors make simplifying assumptions about the GCN model (all ReLU paths are equally likely) and the distribution of the individual nodes' thresholds, theta_j. The attack is evaluated experimentally on three well-known datasets and three different models. The proposed attack outperforms the centrality-based baselines and the baselines of [Ma et al. 2020].

Strengths:
---------

* The paper is well-written and generally easy to follow.
* The connection to influence maximization is interesting.
* The problem of GNN robustness under (realistic) adversarial attacks is important.

Weaknesses:
-----------

* The description of the method and experimental set-up leave a number of open questions (see detailed feedback).
* Despite claiming black-box attacks, the method actually uses the target model's gradients, even for the test set, for the attack.
* The experimental evaluation is rather slim. For example, there is no analysis of the node selection algorithm, e.g. what kind of nodes are selected, no analysis of the sharp differences in performance on the different models, and no analysis of the hyperparameters a and sigma.

Detailed comments:
------------------

I have several concerns about both the method as it is described in the paper as well as the experimental evaluation.


Major points:

Method:
* The authors emphasize the black-box nature of their attack, however the perturbation vector requires access to the respective model's gradient, which violates the black-box setting. This is especially critical since Eq. (10) involves the gradient of the loss of ALL nodes, including the validation and test nodes. So effectively, the attack utilizes the gradient of the loss of the test nodes. Also, it is not stated which value of lambda is used for the perturbation, i.e. what magnitude the perturbation has. The authors mention that this is done because the input features lack semantic meaning; however, they do not mention how potential semantic meaning of the features could be used instead of using the gradients.
* The neighbor weight of alpha_{ij} = 1/|N_i| is not the weight proposed by [Kipf and Welling 2017]. Instead, they propose alpha_{ij} = (d_i+1)^{-0.5} * (d_j+1)^{-0.5}, where d_i is the degree of node i.
* Why is the attack target the misclassification rate on the whole dataset (Eqs. (3,5,6,7)) and not only on the test set? Does this mean that the results in Table 1 are also reported on the whole dataset? If not, why is there a mismatch between the reported misclassification rate and the optimization objective?
* Eq. (4) is unclear/not well defined. As stated above the equation, hat{k}_j is the predicted class of node j after perturbing the set S. However, if S fails to change the predicted class of node j, the numerator and the denominator both become 0, i.e. ill-defined.
* Sec. 4.4: the authors mention that the first approximation leads to the problem becoming "likely to be submodular". Why is it only likely to be submodular and not guaranteed?
* Also Sec. 4.4: According to the authors, the first approximation "integrates out the randomness in data [...]". Where is there any randomness in the data?
* Section 4 shows how the attack can be instantiated for GCN. How is the attack adapted to the other models, i.e. GAT and JKNet?
* The paper just briefly mentions that Eqs. (6) and (7) are used to select the target nodes via a greedy influence maximization approach. A few more sentences on how this will be exactly done would be readers not familiar with inf. max.
* The authors should give more details (proof and/or reference to existing work) regarding the derivation of Corollary 1 and 2.


Experimental evaluation:
* Unrealistic set-up: 60% train data. For semi-supervised node classification, typically we have 10% or less training samples.
* The authors use L=4 layers for the GNNs. However, [Kipf and Welling 2017] report substantially worse performance for L=4 compared to, e.g., L=2. What are the results of the attack when the L used for attacking is different than the L of the GNN?
* The node attributes in the datasets have special constraints, e.g. for Citeseer and Cora the features are binary, and for Pubmed the features are nonnegative. How is this accounted for by the perturbation model?
* How do the choice of sigma and a influence the results?
* Is there any insight into why JKNet and GAT seem to be much more fragile to the attacks? Even for the otherwise very weak random attack GAT's and JKNet's performances drop sharply. Why does this happen?
* There is no analysis into what kind of nodes the proposed algorithm selects; since node selection is the main contribution of this paper, some insight into this would greatly benefit the experimental evaluation.


Minor points:
* p. 1 original from => original form
* p. 6 first paragraph: non-decreasing => non-increasing
* The paper does not cite the correct sources for the dataset.

---

> ### Author Response · Authors · 2020-11-17
> **Response to AnonReviewer4 (Part 1)**
>
> We are very grateful for the detailed review, which helps improve our draft a lot in the updated version. We provide both in-depth discussions and multiple additional experiments in the updated draft to address all the concerns raised by the reviewer. In particular, we would like to highlight that 1) we clarify that the use of gradient information is very mild, and further add a synthetic data experiment with a pure black-box setup; 2) we add sensitivity analysis experiments; 3) an additional experiment on the proposed methods that targets on the test set further improves the attack performance on the test accuracy, thanks to the suggestion by the reviewer.
>
> ---
>
> #### Method
>
> (1) Black-box evaluation
>
> We acknowledge that the current evaluation setup is not a perfect black-box setup. However, we first note that we follow the same evaluation setup as [1] and highlight that the current use of model information is very mild: determining a few number of important features and the binary direction to perturb for each selected feature, only at the **global level** (the same constant perturbation is added to each node to be perturbed). Such mild information should be available if we have more domain knowledge about the task and the features, and then we can move to a full black-box setup.
>
> We added in the updated draft that $\lambda$ is set as 1.
>
> To verify this and further address the concern, we also add *an additional synthetic data experiment* following the recently updated version of [1]. In particular, we generate synthetic node classification data where a couple of important features for the task are known as (model-agnostic) domain knowledge, so that we can construct a constant perturbation vector in the *pure black-box setting*. And we demonstrate that the proposed strategies still outperform the baselines (see Table 3 in the updated draft).
>
> (2) Neighbor weight
>
> Thanks for spotting this mismatch. The D^{-1}A variant of GCN was used in [2] and also works well in practice. We changed the citation at the end of Section 3.1 to [2] in the updated draft to make it precise.
>
> It is also worth noting that, despite the nuance between the analyzed model and the empirically evaluated model (GCN, GAT, JKNet), the derived attack strategies show effective performance in the experiments, which suggests the nuance is not critical for the purpose of developing attack strategies.
>
> (3) Target on test set
>
> First, we clarify that the results in Table 1 are the accuracy obtained on the test set (now explicitly stated in the updated draft). Second, we note that the objectives can be easily adapted to sum over the test set by changing the summation terms.
>
> We agree with the reviewer that targeting on the test set should further improve our proposed strategies in terms of the drop of accuracy on the test set. This is indeed verified by *an additional experiment* we added in the updated draft. By modifying the proposed strategies to target on the test set, the attack performances are consistently improved in all settings (see Table 2 in the Appendix).
>
> (4) Edge case of Eq. (4)
>
> Thanks for spotting it. We have now defined the $\theta_j$ to be $\infty$ when $\hat{k}_j = y_j$ in the updated draft.
>
> (5) Likely to be submodular
>
> The submodularity of the expected objective h(S) relies on the marginal CDFs to be concave. So it is only "likely".
>
> (6) Randomness in the data
>
> The \theta depends on both the model parameters and the data, which we do not have full access to (we do not know the model parameters and the node labels). So we instead treat them as random, and taking expectation over \theta essentially integrates out the randomness.
>
> We elaborated a little bit on this paragraph in the updated draft to address 5 and 6.
>
> (7) Adapt to GAT and JKNet
>
> We note that the derived attack strategies themselves do not rely on the specific GNN models so they can be directly applied to most GNN models, though without theoretical justifications. Nevertheless, the goal of the experiments on GAT and JKNet is to demonstrate that the strategies theoretically derived for GCN can be empirically generalized to various other types of GNN models.
>
> (8) Explanation of the greedy strategy
>
> Added the following explanation in the updated draft: each strategy iteratively selects nodes into the set to be perturbed up to a given size. At each iteration, the node, combining with the existing set, that maximizes Eq. 5 or Eq. 6 will be selected.
>
> (9) Derivation of Corollary 1 and 2
>
> Added the explanation: Corollary 1 and 2 directly follow Proposition~\ref{prop:submodular} given the cumulative distribution functions of the uniform distribution and the normal distribution as well as the fact that they are concave at the positive region.
>
> ---
>
> (To be continued)

---

> > ### Author Response · Authors · 2020-11-17
> > **Response to AnonReviewer4 (Part 2)**
> >
> > #### Experimental evaluation
> >
> > (1) Training split
> >
> > We first note that we are following the training split setup of [1], which itself follows the setup of JKNet [3]. We also note that, as the goal of the experiments is to demonstrate the effectiveness of the attack strategies rather than improving the classification performance, we consider the training split as a nuisance parameter.
> >
> > (2) The parameter L
> >
> > We clarify that the parameter L only refers to the number of random walks used in the attack strategies but does not refer to the number of GNN layers. We used the common choice of number of layers for each model, i.e., 2 layers for GCN and GAT, and 7 layers for JKNet.
> >
> > The fact that the number of random walks used in the attack strategies is different from the number of layers in the GNN models suggests that the proposed strategies do not need to know the exact number of layers of the GNN models to be effective. This phenomenon is also reported in [1] for the baseline GC-RWCS.
> >
> > (3) Feature constraints
> >
> > We used the datasets preprocessed by the Deep Graph Library [4], where the features are already made continuous. So we didn't consider the feature constraints in our experiments.
> >
> > This is indeed an interesting question. However, we consider it out of the scope of the main goal of this work. We added it as future work in our conclusion.
> >
> > (4) Choice of $\sigma$ and $a$
> >
> > We first note that the $\sigma$ and $a$ are fixed as 0.01 in all our experiments and they generalize well across datasets and models. The results are also fairly robust for a range of choices and we added a sensitivity analysis in Figure 4.
> >
> >
> > (5) Performance on GAT and JKNet
> >
> > Unfortunately we have no solid clue so far for this phenomenon. But one guess is that more complicated models tend to overfit the decision boundaries more, and thus are more sensitive to the adversarial attack.
> >
> > (6) Analysis on the nodes being selected
> >
> > This is a good and also hard question! It is not yet clear to us what characteristics at the node level will provide the most effective attack, as otherwise we can directly select nodes based on these characteristics. One insight we do have is that, not surprisingly, the selected nodes by our methods do not have the highest centrality scores, which indicates most existing centrality measures are not good indicators in terms of adversarial attack. We plan to further investigate along this direction in the future work, which may potentially lead to simpler attack strategies.
> >
> > ---
> >
> > #### Minor points
> >
> > We have corrected them in the updated draft. Thanks for catching them!
> >
> > ---
> >
> > #### References
> >
> > [1] Jiaqi Ma, Shuangrui Ding, Qiaozhu Mei. Black-box adversarial attacks on graph neural networks with limited node access. NeurIPS 2020.
> >
> > [2] Will Hamilton, Zhitao Ying, Jure Leskovec. "Inductive representation learning on large graphs." NeurIPS 2017.
> >
> > [3] Keyulu Xu, Chengtao Li, Yonglong Tian, Tomohiro Sonobe, Ken-ichi Kawarabayashi, and Stefanie Jegelka. Representation learning on graphs with jumping knowledge networks. ICML 2018.
> >
> > [4] https://www.dgl.ai/

---

> > > ### Comment · AnonReviewer4 · 2020-11-24
> > > **Response**
> > >
> > > Thank you for the clarifications.
> > >
> > > * My recommendation would be to not use the term "black box" (even if the access to the model's parameters is relatively minor). Further, a slightly more realistic attack setting could be to restrict the gradient access to the attack to the training partition, and then evaluate the results on the test partition.
> > > * Regarding the parameter L: in the current manuscript in Section 3.1 it still says "We assume the GNN f has L layers". So here it still describes the setting where all GNNs have L=4 layers.
> > > * The synthetic experiment is interesting, but needs better explanation. It says the features are generated "randomly from a multivariate normal distribution". What is the mean / covariance matrix here? Further, the description "we assume that the attacker knows a few important features with large corresponding weights in W" is too vague.
> > >
> > > The authors have put in significant effort to address my concerns. However, some of my concerns remain and therefore I update the review score to 5. I recommend the authors to carefully consider the title of the work (i.e. use of the term "black-box") and to re-submit an updated version, since the approach itself is interesting and promising.

---

> > > > ### Author Response · Authors · 2020-11-24
> > > > **Thanks for the suggestions and we've updated the draft accordingly!**
> > > >
> > > > We appreciate the detailed suggestions and we've updated the draft accordingly.
> > > >
> > > > ---
> > > >
> > > > (1)
> > > >
> > > > We changed the term from "black-box" to "near-black-box" per the suggestion by the reviewer.
> > > >
> > > > In addition, we also *added the suggested experiment setup* (restricting to the training partition) and the results are now shown in Table 4 in Appendix A.3 of the updated draft. The results are very similar to those in Table 1, which is not surprising since we only use coarse gradient information.
> > > >
> > > > ---
> > > >
> > > > (2)
> > > >
> > > > We further updated the draft to clarify in Sec. 5.1 that in practice, "the hyper-parameter L does not have to be the same as the number of layers of the  GNN being attacked", and to clearly state in Sec. 5.2 that our experiments used 2-layer GC, 2-layer GAT, and 7-layer JKNet.
> > > >
> > > > ---
> > > >
> > > > (3)
> > > >
> > > > We added more details of the synthetic experiments in our updated draft. Further for your reference, we used the code by [Ma et al. 2020], which is publicly available here: https://github.com/Mark12Ding/GNN-Practical-Attack/blob/main/utils.py#L11

---

### Official Review · AnonReviewer2 · 2020-10-29
**Interesting connection revealed, efficiency claim unsubstantiated**

**Rating:** 6
**Confidence:** 4

**Review:**

This paper introduces a novel connection between adversarial attack on graph neural networks in a restricted black-box setup via node feature perturbation, on the one hand, and the influence maximization problem under the linear threshold model on the same graph, on the other hand. An analysis shows that the objective function of the corresponding IM problem is submodular under assumption, hence the problem admits greedy approximation algorithms as effective black-box attack strategies. Experiments show such attacks are effective compared to baselines in degrading the performance of GNNs in terms of mis-classification rate.

A question that remains unaddressed is whether the analogy to the IM problem under the LT model extends to other properties of the problem: under the uniform distribution of threshold θ, the diffusion of influence under the LT is equivalent to a diffusion whereby each node picks at most incoming edge to be active with probability corresponding to the incident edge's weight. This paper would be stronger if it addressed the question of whether this property also extends to the analogy, apart from the submodularity property.

Under the current analysis, two greedy algorithms are proposed for two different objectives, yet there is no analysis of their complexity and no runtime results on their efficiency. The algorithms are called efficient, but no evidence is provided to that effect. That is a weak point of the paper.

---

> ### Author Response · Authors · 2020-11-17
> **Response to AnonReivewer2**
>
> We thank the reviewer for the recognition as well as the questions. And we answer your specific questions below.
>
> ---
>
> (1) Analogy to IM.
>
> Under the assumptions in Corollary 1, the problem reduces to the classic IM problem under LT where $\theta$ has uniform distributions. In this case, we believe those properties of the general IM problem under LT do extend to our specific problem. We added this comment under the Corollary 1 and 2 in the updated draft.
>
> ---
>
> (2) Complexity.
>
> In practice, both the proposed strategies and most baselines are very fast. We call them "efficient" in the sense that the solutions are efficient compared to solving the original combinatorial optimization problem.
>
> A more detailed analysis is shown here. Both the proposed strategies and RWCS/GC-RWCS need to first compute an L-step (L=4 in this paper) random walk probability matrix B through sparse matrix multiplication, which is O(NE), where N is the number of nodes and E is the number of edges. This step is efficient for sparse real-world graphs. For RWCS, one only needs to have a column sum of B, then take argmax over the sums, which in total is O(N^2). For the proposed strategies and GC-RWCS, we need to do some minor updates on B and do the column sum for each node to be selected. And the operation in each step is still O(N^2). To greedily select r nodes, we would have a time complexity of O(rN^2). Further, in the case when we only target at the test set of, say, S nodes, all the O(N^2) will be reduced to O(NS). We also note that the most of the O(N^2) operations can be made parallel and exploit the sparsity of the matrix B.

---

### Decision · Program_Chairs · 2021-01-07
**Final Decision**

**Decision:**

Reject

**Comment:**

This paper relates the problem of influence maximization and adversarial attacks on GCNs.
The paper, and its formulation and assumptions stirred up quite a discussion among the reviewers and the authors. I do appreciate the thorough rebuttal that the authors provided, and the reviewers did take it into account (and revised their scores).
However, all in all, I am afraid that there are just a few too many concerns with this paper.
If the authors take the reviews to heart, they should be able to improve the manuscript and submit a stronger and improved version to the next conference.